# Biochemical analyses of cystatin-C dimers and cathepsin-B reveals a trypsin-driven feedback mechanism in acute pancreatitis

Jana Marielle Modenbach[1,7], Christina Möller [2,7], Saeedeh Asgarbeik [1,7], Norman Geist [3], Niklas Rimkus [2], Mark Dörr [2], Hannes Wolfgramm [4], Leif Steil [4], Anne Susemihl [3,5], Leonie Graf [6], Ole Schmöker [6], Dominique Böttcher[2], Elke Hammer [4], Juliane Glaubitz[1], Michael Lammers [6], Mihaela Delcea[3], Uwe Völker [4], Ali Alexander Aghdassi[1], Markus M. Lerch[1], Frank Ulrich Weiss [1], Uwe T. Bornscheuer [2,8] ✉ & Matthias Sendler [1,8] ✉

Acute pancreatitis (AP) is characterised by self-digestion of the pancreas by its own proteases. This pathophysiological initiating event in AP occurs inside pancreatic acinar cells where intrapancreatic trypsinogen becomes prematurely activated by cathepsin B (CTSB), and induces the digestive protease cascade, while cathepsin L (CTSL) degrades trypsin and trypsinogen and therefore prevents the development of AP. These proteases are located in the secretory compartment of acinar cells together with cystatin C (CST3), an endogenous inhibitor of CTSB and CTSL. The results are based on detailed biochemical analysis, site-directed mutagenesis and molecular dynamics simulations in combination with an experimental disease model of AP using CST3 deficient mice. This identifies that CST3 is a critical regulator of CTSB and CTSL activity during AP. CST3 deficient mice show a higher intracellular CTSB activity resulting in elevated trypsinogen activation accompanied by an increased disease severity. This reveals that CST3 can be cleaved by trypsin disabling the inhibition of CTSB, but not of CTSL. Furthermore, dimerised CST3 enhances the CTSB activity by binding to an allosteric pocket specific to the CTSB structure. CST3 shifts from an inhibitor to an activator of CTSB and therefore fuels the intrapancreatic protease cascade during the onset of AP.

Acute pancreatitis (AP) is a common and potentially lethal gastro-intestinal disease. With an incidence of 13–45/100.000 per year worldwide[1] AP is one of the main causes for hospitalisation among diseases of the gastrointestinal tract, which results in high costs for the health care systems[2]. AP starts within pancreatic acinar cells and triggers an immune response which determines the course of the disease[3–5]. To the present day, no causal treatment for AP exists[5].

[1]Department of Medicine A, University Medicine Greifswald, Greifswald, Germany. [2]Department of Biotechnology and Enzyme Catalysis, Institute of Biochemistry, University of Greifswald, Greifswald, Germany. [3]Department of Biophysical Chemistry, Institute of Biochemistry, University of Greifswald, Greifswald, Germany. [4]Department of Functional Genomics, Interfaculty Institute for Genetics and Functional Genomics, University Medicine Greifswald, Greifswald, Germany. [5]Department of Medicine C, University Medicine Greifswald, Greifswald, Germany. [6]Department of Synthetic and Structural Biochemistry, Institute of Biochemistry, University of Greifswald, Greifswald, Germany. [7]These authors contributed equally: Jana Marielle Modenbach, Christina Möller, Saeedeh Asgarbeik. [8]These authors jointly supervised this work: Uwe Bornscheuer, Matthias Sendler. ✉e-mail: uwe.bornscheuer@uni-greifswald.de; matthias.sendler@uni-greifswald.de

Animal experiments, as well as clinical and genetic studies, suggest that the serine protease trypsin plays a key role in the pathogenesis of AP[6–10]. Under physiological conditions, its catalytically inactive precursor trypsinogen is sorted into zymogen granules and becomes activated after secretion into the duodenum by the brush border enzyme enterokinase. Trypsin, in turn, proteolytically activates the other zymogens, including chymotrypsinogens, proelastases, and procarboxypeptidases. Premature activation of trypsinogen within pancreatic acinar cells is a critical event for the onset of AP[6]. In the absence of enterokinase, the lysosomal hydrolase cathepsin B (CTSB) is able to activate trypsinogen to active trypsin[7,11]. Animal studies have shown that CTSB is the main intrapancreatic activator of trypsinogen and that the genetic deletion of *Ctsb* prevents trypsinogen activation completely[7,11,12]. The cellular compartment in which this activation takes place is still under debate[13]. A co-localisation of both enzymes in the same compartment is a mandatory requirement for trypsinogen activation via CTSB. Two main theories of co-localisation were discussed: (1) a fusion of lysosomes and zymogen granules[14], or (2) a missorting of CTSB into zymogen granules[15]. Animal studies investigating subcellular fractions of the pancreas suggest that activation of trypsinogen takes place in the heavy compartment containing zymogen granules[11]. Interestingly, CTSB, the trypsinogen-activating enzyme, is present in the secretory compartment[16], but under physiological conditions, this co-localisation does not result in premature activation of trypsinogen. Besides CTSB, the trypsin and trypsinogen-degrading enzyme cathepsin L (CTSL) is located in the same compartment[11,17]. This suggests a regulatory mechanism that prevents CTSB-mediated trypsinogen activation and controls the activity of the lysosomal enzymes CTSB and CTSL. Cysteine proteases such as CTSB and CTSL are inhibited by the endogenous inhibitor cystatin C (CST3)[18]. CST3 is a small non-glycosylated protein that is present in nearly all cells and belongs to the type 2 of secreted cystatins[18].

In this study, we investigated the role of CST3 in the onset of AP. We found that the human and mouse pancreatic CST3 undergoes sorting into the secretory compartment of acinar cells, where it is the central regulator of CTSB and CTSL activity. CST3 deficient mice (*Cst3*–/–) showed a higher intracellular trypsin activation and developed more severe disease during caerulein-induced AP. Furthermore, we identified a dimerisation mechanism of CST3 which plays a central role in the control of CTSB and CTSL. These in-depth experiments provide evidence that CST3 has a protective function in the pancreas, by suppressing CTSB activity in the trypsinogen-containing secretory compartment. Under conditions of AP this protective function is reversed and, instead, fuels the activation of the protease cascade.

## Results

### Cystatin C is localised in the secretory compartment of pancreatic acinar cells

Immunofluorescence labelling of CST3 in mouse pancreas sections showed an intracellular localisation within the secretory compartment, co-localised with α-amylase (Fig. 1A). Subcellular fractionation of murine pancreatic tissue was performed from untreated control animals and animals 1 h after induction of pancreatitis by stimulation with caerulein. Western blot analysis of subcellular fractions gave evidence that CST3 is localised in organelles of the heavy fraction which we identified as zymogen granules (ZG) by the marker syncollin (Fig. 1B). The lysosomal fraction (Lys) was identified by LIMP-2 whereas cytosolic fraction (Cyt) was marked by GAPDH. In untreated control mice, the mature form of CTSB could be detected in the lysosomal, but also in the zymogen granule fraction where it seemed to accumulate 1 h after induction of AP together with its inhibitor CST3. CTSL, another cathepsin that is target of CST3 showed the same expression profile. Enzyme activity measurements within the subcellular fractions showed that the majority of CTSB activity is localised to the lysosomal compartment in untreated controls but redistributed into the zymogen

granule fraction 1 h after onset of pancreatitis (Fig. 1C). In untreated mice the major activity of CTSL, similar to CTSB, was located in the lysosomal fraction but did not increase in the zymogen granule fraction 1 h after induction of disease (Fig. 1D). Measurement of chymotrypsin and trypsin activities showed a strong activation within the zymogen granule fraction 1 h after caerulein treatment (Fig. 1E, F). We also co-labelled CST3 together with α-amylase in human pancreatic tissue sections (Fig. 1G) and observed a co-localisation comparable to the murine pancreas. Further evidence for the presence of CST3 in the secretory pathway was provided by Western blot detection of CST3 in pancreatic juice samples from patients, together with α-amylase and CTSB (Fig. 1H). In 18 additional samples of human pancreatic juice we could demonstrate a negative correlation of the presence of CST3 with CTSB and trypsin activity (Supplementary Fig. 1A–F). Animal data as well as investigations of human samples suggest that, during pancreatitis, the lysosomal cysteine protease inhibitor CST3 undergoes redistribution into the zymogen-containing secretory compartment together with CTSB and CTSL. These observations raised the question why CTSB became significantly more active in zymogen granules after induction of pancreatitis despite the presence of CST3, whereas CTSL activity was not significantly influenced.

### Protease activation in cystatin C deficient mice

We next investigated the impact of missing CST3 on CTSB activity in CST3 deficient mice (*Cst3*–/–). By Western blot analysis of pancreatic tissue, we verified the absence of CST3 in *Cst3*–/– mice (Fig. 2A). To figure out if enzyme content was different between mice strains, we measured trypsin and chymotrypsin activity after enterokinase activation, to evaluate the total amount of trypsinogen. Pancreatic enzyme content of trypsinogen, chymotrypsinogen and amylase was similar in pancreas homogenates between untreated wild type and *Cst3*–/– mice (Fig. 2B). Labelling of pancreatic amylase and CTSB showed also no differences of pancreatic architecture and CTSB content between wild type and Cst3–/– mice (Fig. 2C). In the next step we isolated pancreatic acini from wild-type and *Cst3*–/– mice, and measured protease activities of CTSB, CTSL, trypsin and chymotrypsin in cell homogenates (Fig. 2D). Following 0.001 mM Cholecystokinin (CCK) stimulation for 30 min acini of *Cst3*–/– mice showed a significantly higher protease activation of cysteine proteases CTSB and CTSL as well as of pancreatic serine proteases trypsin and chymotrypsin. Enzymatic activities were measured under optimised pH conditions (pH 4 for CTSL, pH 5.5 for CTSB and pH 8.0 for trypsin and chymotrypsin) for the respective enzyme, and in cell homogenates, in which subcellular localisation changes can be disregarded. For this reason, we measured enzyme activities in living acinar cells (Fig. 2E) and also observed in *Cst3*–/– acini significantly higher CTSB activities at 20 and 40 min after CCK-stimulation compared to acini from wild-type mice. In line with the previous findings, we also observed an increased activity of trypsin. Interestingly, propidium iodide (PI) uptake, as a marker of necrotic cell death, was not elevated. To find out in which subcellular compartment this increased activity was located we prepared subcellular fractions of pancreatic tissue from control and 1 h pancreatitis animals and measured enzyme activities of CTSB, CTSL, trypsin, and chymotrypsin in these subcellular fractions (Fig. 2F, G). In the zymogen fraction from *Cst3*–/– animals we observed significantly increased activities of CTSB and CTSL during pancreatitis, in comparison to control animals (Fig. 2F). In parallel, increased activation of pancreatic proteases like trypsin and chymotrypsin could be observed in the ZG fraction of *Cst3*–/– mice (Fig. 2G). These findings confirm a protective role of CST3 by inhibiting CTSB and CTSL activity in the zymogen containing fraction, which prevents the activation/degradation of trypsinogen. Our next aim was to elucidate the molecular mechanisms underlying our observation that under conditions of acute pancreatitis, CST3 lost its capacity to inhibit CTSB and CTSL-supporting disease progression.

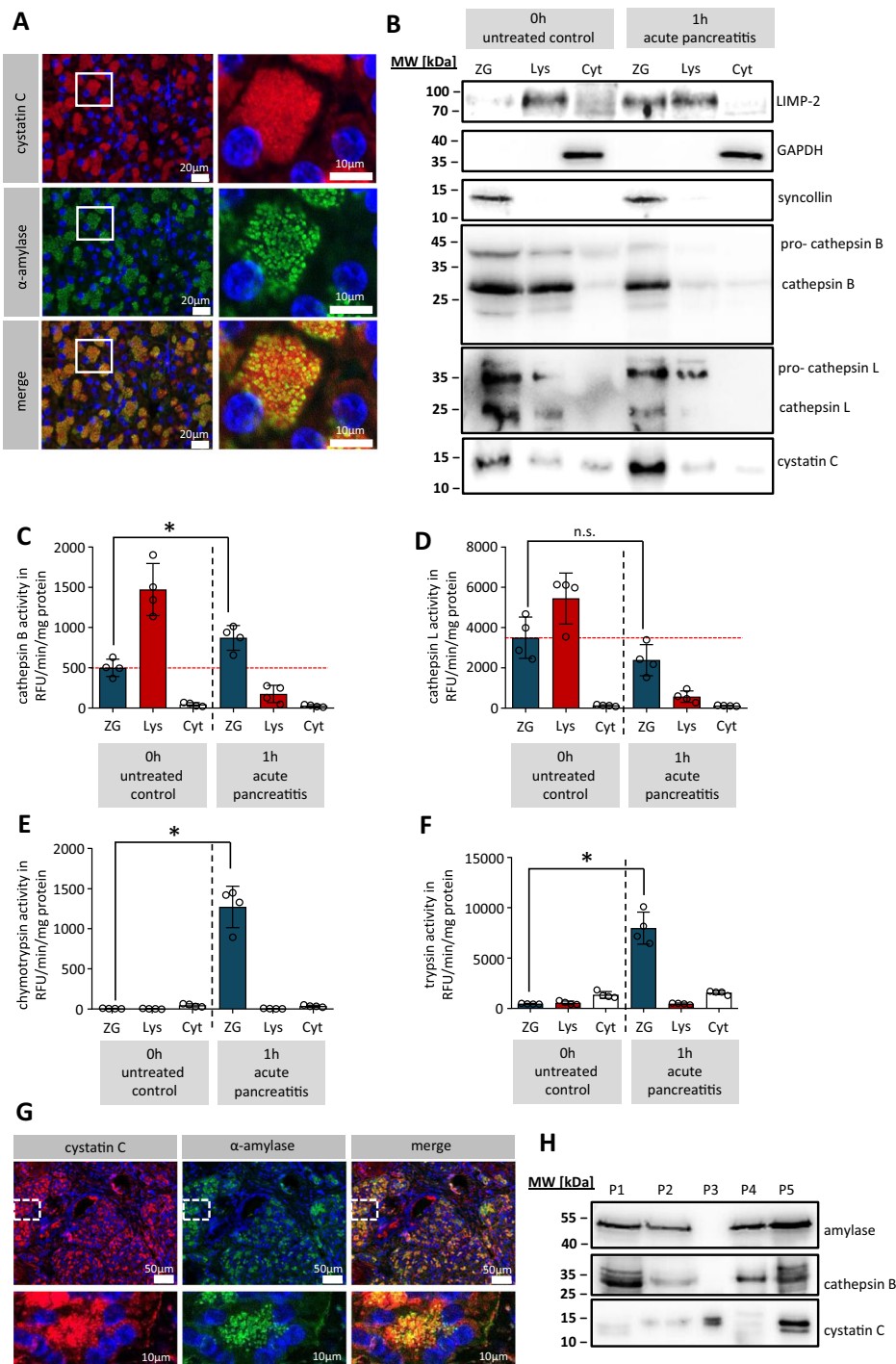

**Fig. 1 | Cystatin C expression in the secretory compartment of pancreatic acinar cells. A** Representative immunofluorescent labelling of cystatin C (red) and pancreatic α-amylase (green) in sections of mouse pancreatic tissue, the experiment was repeated at least three times independently with a similar result (the scale bar represents 20 μm and 10 μm). **B** Western blot analysis of subcellular fractions (ZG zymogen granules, Lys lysosomal fraction, and Cyt cytosolic fraction) from tissue lysates of mouse pancreas, taken from untreated control mice and animals 1 h after induction of pancreatitis by caerulein, the experiment was repeated four times independently with a similar result. Enzyme activity measurements of CTSB ($p = 0.0073$) (**C**), CTSL (**D**), chymotrypsin ($p < 0.0001$) (**E**) and trypsin ($p < 0.0001$) (**F**) in mouse subcellular fractions ($n = 4$). **G** Immunofluorescent labelling of cystatin C (red) and pancreatic α-amylase (green) in sections of human pancreatic tissue the experiment was repeated three times independently with a similar result (the scale bar represents 50 μm and 10 μm). **H** Western blot analysis of five different human pancreatic fluid samples. **C–F** show four independent experiments; significance was calculated by two-tailed Student *t*-test for independent samples. Results are shown as mean ± SD. Significance levels of $p < 0.05$ are marked by an asterisk. Source data are provided as Source Data file.

**Regulation of protease activity by cystatin C during pancreatitis**

It is known that the activity of lysosomal enzymes such as CTSB or CTSL is influenced by the pH and that the initiation of protease activation starts in an acidic compartment[11,19]. In the next step, we

therefore investigated how AP-induction affects the activity of CTSB in the zymogen granule fraction under specific pH conditions. At 1 h after the onset of pancreatitis we observed an increased CTSB activity under mild acidic to neutral pH range (6–7) (Fig. 3A). We analysed whether

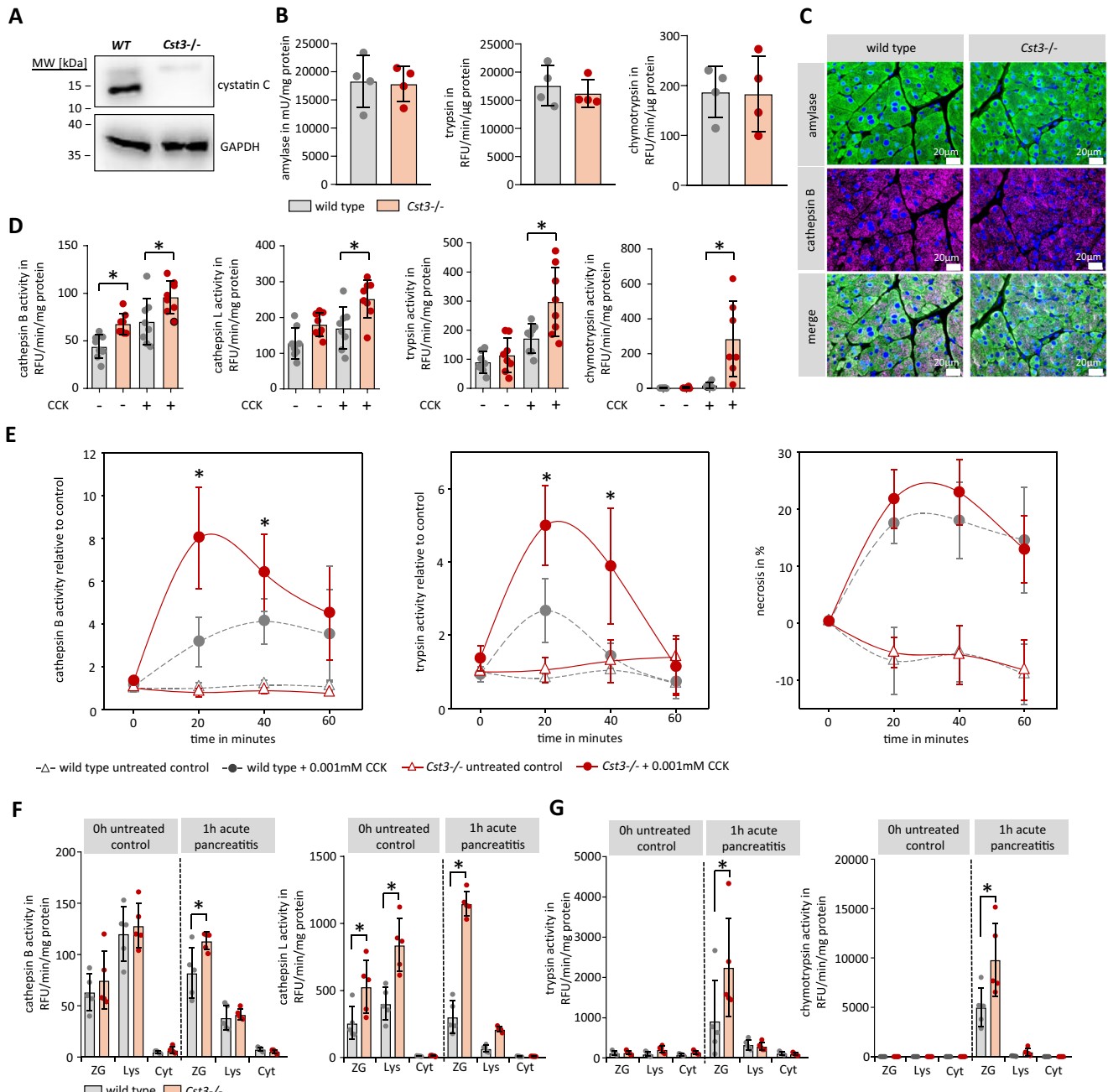

**Fig. 2 | Protease activation in cystatin C deficient mice. A** Western blot analysis of cystatin C in the pancreas of wild-type (WT) and *Cst3−/−* mice, GAPDH reference as loading control. **B** Trypsin and chymotrypsin activities were measured after enterokinase activation as well as amylase content in the pancreas of WT and *Cst3−/−* mice (*n* = 4 biological replicates). **C** Immunofluorescent labelling of pancreatic α-amylase (green) and CTSB (violet) in pancreas sections of wild type and *Cst3−/−* mice (scale bar represents 20 μm). **D** Protease activities were measured using fluorochrome substrates for CTSB (*n* = 8 biological replicates, without CCK, *p* = 0.0011, +CCK *p* = 0.0285), CTSL (*n* = 8 biological replicates, +CCK *p* = 0.0114), trypsin (*n* = 8 biological replicates, +CCK *p* = 0.0145), and chymotrypsin (*n* = 7 biological replicates, +CCK *p* = 0.0071) in homogenates of isolated acini from wild-type and *Cst3−/−* mice ± stimulation for 30 min with 0.001 mM CCK. **E** Protease activities of cathepsin B (*n* = 5 biological replicates, 20 min *p* = 0.0034, 40 min *p* = 0.0399), trypsin (*n* = 5 biological replicates, 20 min *p* = 0.0057, 40 min

*p* = 0.0092) and propidium iodide (*n* = 10 biological replicates) uptake were measured in freshly prepared living acinar cells of wild-type and *Cst3−/−* mice. **F** Activities of cathepsin B (1 h ZG, *p* = 0.0275) and cathepsin L (0 h ZG, *p* = 0.032, 0 h Lys *p* = 0.003, 1 h ZG *p* < 0.001) were measured by fluorochrome substrates in subcellular fractions (ZG zymogen granules, Lys lysosomal fraction, and Cyt cytosolic fraction) of wild-type and *Cst3−/−* untreated control mice and 1 h after induction of pancreatitis (*n* = 5 biological replicates). **G** Activity measurements of trypsin (1 h ZG, one-tailed students *t*-test *p* = 0.0492) and chymotrypsin (1 h ZG, *p* = 0.0336) in the same subcellular fractions (*n* = 5 biological replicates). Data represent five or more independent experiments; significance was calculated by two-tailed Student *t*-test for independent samples unless otherwise mentioned. Results are shown as mean ± SD. Significance levels of *p* < 0.05 are marked by an asterisk. Source data are provided as Source Data file.

this pH-dependent activity shift correlates with the presence of CST3 in the secretory compartment and performed the same experiments in *Cst3−/−* mice. Here we did not observe a comparable change and the pH range of CTSB activity in zymogen granules was similar to wild-type

pancreatitis mice (Fig. 3B). These observations suggest a loss of the inhibitory capacity of CST3 on CTSB after induction of pancreatitis. Interestingly, we did not observe this pH effect on CTSL activity after induction of pancreatitis in wild-type mice (Fig. 3C). While in *Cst3−/−*

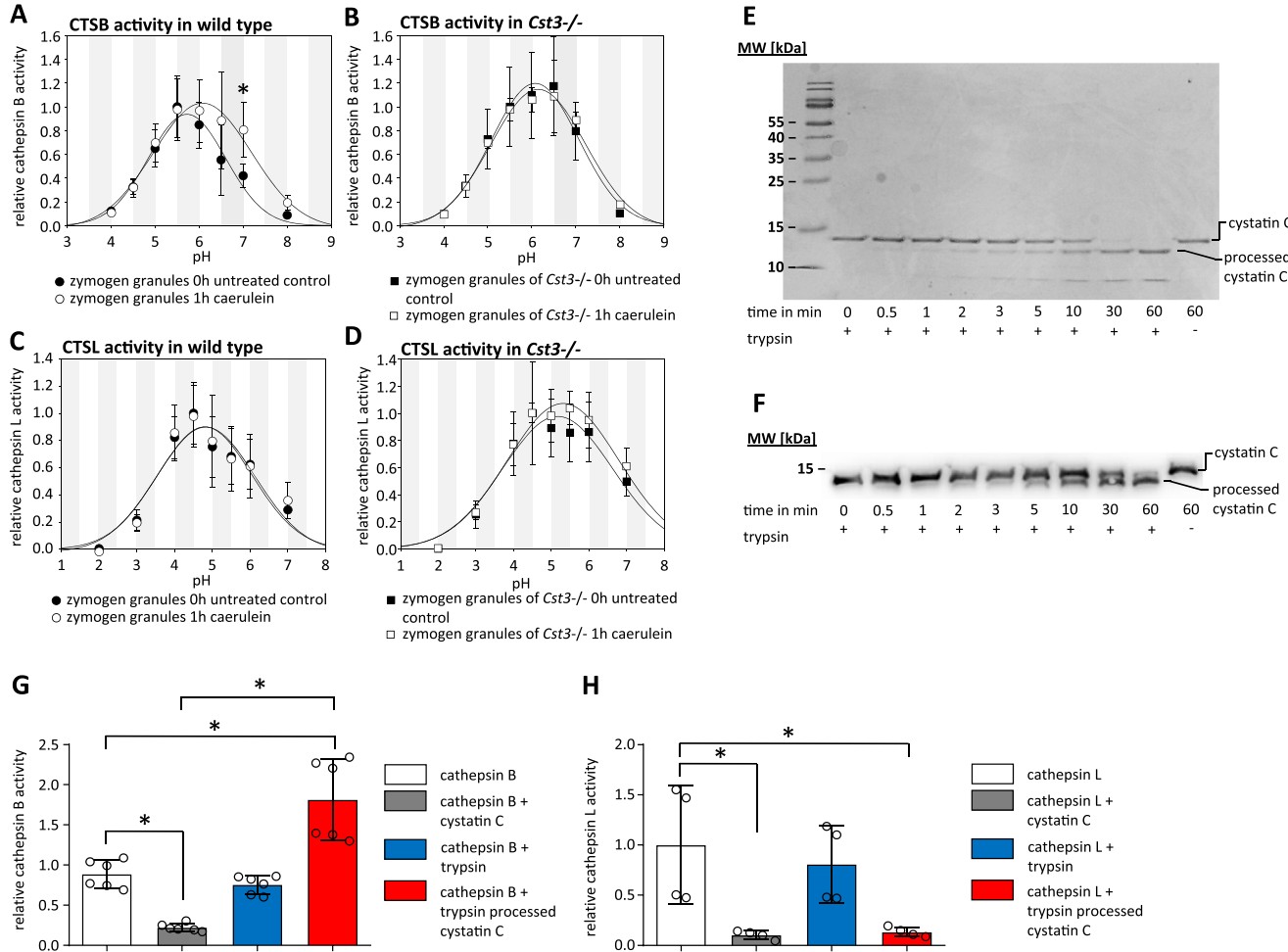

**Fig. 3 | Regulation of protease activity by cystatin C during pancreatitis.** Measurement of CTSB activity at different pH conditions (pH 3–9) in zymogen granule fractions of wild-type mice (pH 7, p = 0.0169) (**A**) and *Cst3*–/– mice (**B**) before and after induction of pancreatitis (*n* = 4 biological replicates). Measurement of CTSL activity under different pH conditions in ZG-fraction of wild-type mice (**C**) and *Cst3*–/– mice (**D**) before and after induction of pancreatitis (*n* = 4 biological replicates). **E** Coomassie staining illustrates the time-dependent cleavage of recombinant cystatin C by trypsin over a time period of 60 min, the experiment was repeated three times independently with a similar result. **F** Western blot analysis of the time-dependent cleavage of cystatin C by trypsin, the experiment was repeated three times independently with the similar result. **G** Measurement of CTSB activity

at pH 5.5 in the presence of cystatin C, trypsin and preincubated cystatin C with trypsin, (*n* = 6 biological replicates, CTSB vs. CTSB + CST3 p = 0.0025, CTSB vs. CTSB + Try/CST3 p < 0.0001, CTSB + Try + Try/CST3 p < 0.0001) (**H**) measurement of CTSL activity at pH 4.0 in the presence of cystatin C, trypsin and preincubated cystatin C with trypsin (*n* = 4 biological replicates, CTSL vs. CTSL + CST3 p = 0.0225, CTSL vs. CTSL + Try/CST3 p = 0.0282). Data represent four independent experiments; significance was calculated by two-tailed Student *t*-test for independent samples or one-way Anova followed Dunn–Šidák correction for multiple testing. Results are shown as mean ± SD. Significance levels of p < 0.05 are marked by an asterisk. Source data are provided as Source Data file.

mice no pH-dependent changes could be observed, the CTSL activity optimum was measured at a slightly more acidic pH of 4–4.5 (Fig. 3D), whereas CTSB showed the highest activity at pH 5.5 in the ZG fraction (Supplementary Fig. 2a). Taken together these findings suggest processing of CST3 in the zymogen granule fraction during pancreatitis which affects its inhibitory capacity for CTSB but not for CTSL. As Trypsinogen undergoes activation in the zymogen granule fraction, we investigated the capacity of trypsin to proteolytically process CST3. Co-incubation of recombinant CST3 with trypsin at 37 °C for 60 min caused cleavage of CST3, as shown in Coomassie-stained gels (Fig. 3E) and in Western blot analysis (Fig. 3F). Next, we investigated the inhibitory effect of processed CST3 on CTSB and CTSL activities. While CTSB activity was clearly reduced in the presence of CST3, it was increased in the presence of processed CST3 (Fig. 3G). More unexpected was the observation, that the processed CST3 was able to inhibit CTSL in the same manner as physiological CST3 (Fig. 3H). Apparently, the cleavage of CST3 by trypsin alters the balance of trypsinogen activating CTSB and trypsin degrading CTSL in

favour of trypsin activation. Now we were interested if we could observe a degradation of CST3 in the ZG-Fraction of mice 1 h after onset of pancreatitis. Because of the missing His-Tag, we performed 17.5% SDS PAGE, to visualise size shift. Densitometry of CST3 western blots showed a clear degradation of CST3 and the occurrence of a processed form of CST3 (Supplementary Fig. 2b, c). The ratio of uncut CST3 to processed CST3 changed significantly in favour of the processed variant after induction of pancreatitis. Multiple proteases become activated during pancreatitis which may cleave CST3. For this reason, we investigated the impact of another serine protease on CST3. In contrast to trypsin, chymotrypsin did not show any specific cleavage of cystatin C, nor did the inhibitory capacity of cystatin C change after chymotrypsin co-incubation (Supplementary Fig. 2d–f).

**Trypsin-mediated cleavage of cystatin C**

In order to investigate how trypsin-processed CST3 can increase CTSB activity while it still inhibits CTSL, we aimed to identify the cleavage site of CST3 by trypsin. Western blot analysis, using anti-His-tagged

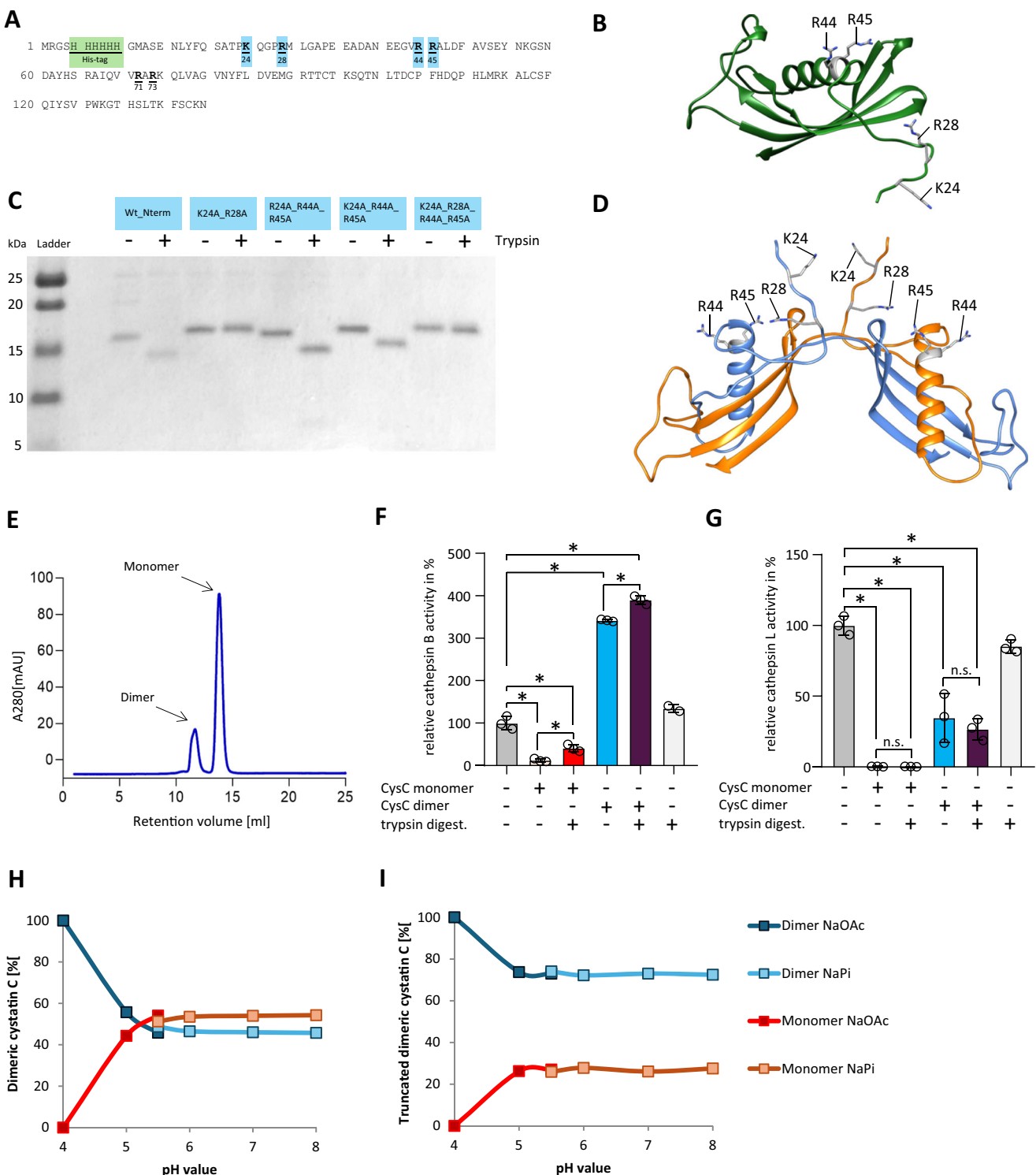

recombinant murine CST3, gave evidence that part of the His-tagged N-terminus is cleaved off. Therefore, no protein band for cleaved cystatin C is detectable (Supplementary Fig. 2g). In the next step, a cleavage site analysis was performed by LC-ESI-MS/MS analysis of the cleaved and the complete form of CST3, which revealed the possible trypsin cleavage sites R28, R44, and R45 in the CST3 protein sequence (Fig. 4A, B). Based on LC-ESI-MS/MS and Western blot analysis, the cleavage occurs close to the N-terminus. In order to eliminate all potential cleavage sites for trypsin, the cleavage sites K24, R28, R44 and R45 were mutated by site-directed mutagenesis to replace arginine at positions 28, 44 and 45 and

lysine 24 by alanine. However, individual amino acid replacements and a double mutation (R44A_R45A) did not prevent the processing by trypsin (Supplementary Fig. 2h). This observation suggested that several cleavage sites for trypsin must be present in CST3. Only multiple amino acid substitutions by mutagenesis of the double mutant K24A_R28A and the quadruple mutant K24A_R28A_R44A_R45A did prevent cleavage by trypsin (Fig. 4C). These results gave evidence that CST3 possesses two trypsin cleavage sites at residues K24 and R28. The CST3 molecule is stabilised by two disulphide bonds and the protein stability is not affected by trypsin cleavage (Supplementary Fig. 3).

**Fig. 4 | Trypsin-mediated cleavage of cystatin C and its oligomerization state.**
**A** Amino acid sequence of murine cystatin C expressed in *E.coli* SHuffle T7 Express with highlighted His-tag for purification (green) and potential trypsin-cleavage sites (yellow). **B** AlphaFold2 structure of murine cystatin C (AlphaFoldDB: AF-P21460 [https://www.uniprot.org/uniprotkb/P21460/entry]) with potential trypsin cleavage sites. **C** SDS-PAGE analysis of cystatin C wild-type and mutants where several possible cleavage sites were substituted with alanine residues. Shown are the purified proteins not treated with trypsin (−) and treated with trypsin (+), the experiment was repeated three times independently with a similar result.
**D** AlphaFold2 structure of murine cystatin C dimer with coloured cystatin C chains in orange and blue and annotated potential cleavage sites. **E** Chromatogram of the separation of cystatin C monomer and dimer fractions by size exclusion chromatography. Activity tests of CTSB at pH 5.5 (con vs. monomer $p < 0.0001$, con vs. monomer + try $p < 0.0001$, con vs. dimer $p < 0.0001$, con vs dimer + try $p < 0.0001$,

monomer vs. monomer + try $p = 0.0173$, dimer vs. dimer + try $p = 0.0003$, $n = 3$ biological replicates) (**F**) and CTSL at pH 4.0 (con vs monomer $p < 0.0001$, con vs monomer + try $p < 0.0001$, con vs. dimer $p < 0.0001$, con vs. dimer + try $p < 0.0001$, $n = 3$ biological replicates) (**G**) with cystatin C monomer and dimer and trypsin-mediated cleaved cystatin C monomer and dimer. Comparison of pH-related dimerisation of full-length cystatin C (**H**) and trypsin-cleaved cystatin C (**I**). Shown are the dimer/monomer ratios obtained in sodium acetate buffer pH 4.0–5.5 (NaOAc, dark blue and red, respectively) and the dimer/monomer ratios obtained in sodium phosphate buffer pH 5.5–8.0 (NaPi, light blue and orange, respectively). Data represent three independent experiments; significance was calculated by one-way Anova followed Dunn–Šidák correction for multiple testing. Results are shown as mean ± SD. Significance levels of $p < 0.05$ are marked by an asterisk. Source data are provided as Source Data file.

Next, we were interested to analyse whether the cleaved CST3 fragments were able to increase the activity of CTSB (Fig. 3G). It is known that during pancreatitis the subcellular compartment in which trypsinogen undergoes activation by CTSB becomes acidified[19]. Furthermore, it is also known that CST3 can dimerise, especially under acidic conditions, and thereby loses its inhibitory capacity[20]. CST3 refolds in a very tight symmetric dimer while retaining the secondary structure of the monomeric CST3 forms but swaps the helical domains between two monomers (Fig. 4D)[21]. We isolated monomer and dimer fractions of CST3 by size exclusion chromatography (Fig. 4E) and found that approximately 75% of the recombinant CST3 was present as a monomer, while the rest was present as a dimer. After isolation, we digested monomer and dimer fractions with trypsin and investigated how the full-length and processed CST3 monomer and dimer fractions affect CTSB and CTSL activities (Fig. 4F, G). CST3 monomers (mCST3) inhibited CTSB activity as expected, but following trypsin cleavage mCST3 significantly lost its inhibitory capacity on CTSB. CST3 dimers (dCST3), on the other hand significantly increased the activity of CTSB about three-fold (Fig. 4F). Trypsin processing of the dimers further increased the CTSB activity. While mCST3 inhibited CTSL completely, trypsin cleavage of CST3 did not affect its inhibitory capacity towards CTSL. (Fig. 4G). dCST3 still inhibit most of the CTSL activity and the processing by trypsin did not change this. We expect, however, that this inhibition is caused by a small fraction of monomeric CST3 which is still present after separating the fractions or may be recovering due to the structural equilibrium between monomers and dimers in CST3.

We further investigated the pH-dependent dimerisation of cystatin C and could show that at pH 4.0 100% of cystatin C was present as a dimer. Between pH 5.0 and 5.5 we observed a switch in the ratio of predominant dimeric to more monomeric forms of cystatin C, which persisted at higher pH values (Fig. 4H, Supplementary Fig. 4). These results fit to the previously reported observation that CST3 forms more dimers under acidic conditions[20]. Moreover, we analysed if the trypsin-mediated cystatin C truncation also influences the monomer/dimer ratio. We found that at pH 4.0, 100 % dimer was present as it was also shown for the full-length CST3. However, dimers remained as the predominant (~70%) form at pH values from 5.5 to 8.0 (Fig. 4I, Supplementary Fig. 5). We conclude that the trypsin-mediated truncation of cystatin C, indeed, further favours the dimerisation of CST3.

Overall, these results suggest that pancreatitis-induced changes of the monomer/dimer ratio of CST3 and the digestion by activated trypsin causes conditions in which CTSL activity is still inhibited by mCST3 while dCST3 strongly enhances CTSB activity and therefore fuels the protease activation cascade of the disease onset.

## The effect of processed CST3 on CTSB and CTSL
During our investigations of possible cleavage sites of CST3 by trypsin, molecular dynamics (MD) simulations were conducted to explain the raised activity of CTSB. Extensive structural ensembles of various molecular systems were collected using enhanced sampling MD

technique TIGER2h$_{PE}$[22] (Supplementary Table 1). In CTSB a special occluding loop covers the access to the active site and this auto-inhibition mechanism prevents substrate binding and proteolysis[23]. By simulating the isolated CTSB molecule, we established a benchmark for the thermodynamic balance between its open and closed states (Fig. 5A, B). This determination was guided by measuring the centre-of-mass distance between the active site residues (C108, H278, N298) and the occluding loop (185–200). At a chosen distance threshold of ≤16.5 Å that describes the closed state, CTSB remains closed approximately 70 to 80% of the time (Fig. 5C). Subsequently, we were interested in the inhibitory complex of wild-type CTSB and CST3, assuming that the interaction of trypsin-cleaved CST3 fragments could stabilise the occluding loop in an open position. The major binding configuration found was analogous to similar proteins with known crystal structures (e.g. papain, stefin B, PDB 1STF)[24] and showed that dimeric CST3 cannot bind in the same inhibiting position as the monomer, due to the helical domains swap, which extends one of the previous binding turn regions beyond steric tolerance (Fig. 5D). Monomeric CST3 bound within the active site (Supplementary Fig. 6a–f), also engaging with the occluding loop, which had shifted slightly from its usual position to make room for CTS3. The tryptic cleavage sites K24 and R28 revealed to be tightly locked to CTSB's surface. We also investigated the inhibitory complex of CST3 with CTSL (Fig. 5E). The major binding position did not strongly involve the N-terminal region, especially K24. However, we found one smaller structural cluster, similar to CTSB binding, but swapped binding positions for the trypsin cleavage sites K24 and R28 (Supplementary Fig. 7a–e). Trypsin-mediated cleavage may therefore have a stronger effect on the inhibition complex with CTSB compared to CTSL which fits our observed activity measurements. If this effect is due to the lack of N-terminal binding or a consequence of secondary effects like initial recruitment, needs further elucidation.

We next explored potential binding interactions between CTSB and tryptic cleavage products of CST3. To achieve this, we conducted MD simulations starting with a monomeric CST3 fragment, cleaved at position R71, positioned on CTSB's rear side, away from the active site. This placement was selected to minimise bias toward regions near the active centre and promote broader sampling. Unexpectedly, the simulations revealed an energetically favourable cluster of CST3 conformations binding to an unknown pocket on CTSB's rear side. (Fig. 6A and Supplementary Fig. 8a–d). This interaction, which involved CST3 residues as R73 and K74 that are usually occluded by its own N-terminal region (Supplementary Fig. 12a) shifted the occluding loop position and caused a pronounced conformational opening of CTSB's active site. We were able to reproduce this binding configuration for several assumed CST3 constructs differing in their N-terminal truncation, up to the final dimeric CST3 truncated at position R28 (Fig. 6A and Supplementary Figs. 8–11). We measured the average distance of the active site and the occluding loop dependent on the presence of CST3. This revealed the average distance of the active site and the occluding

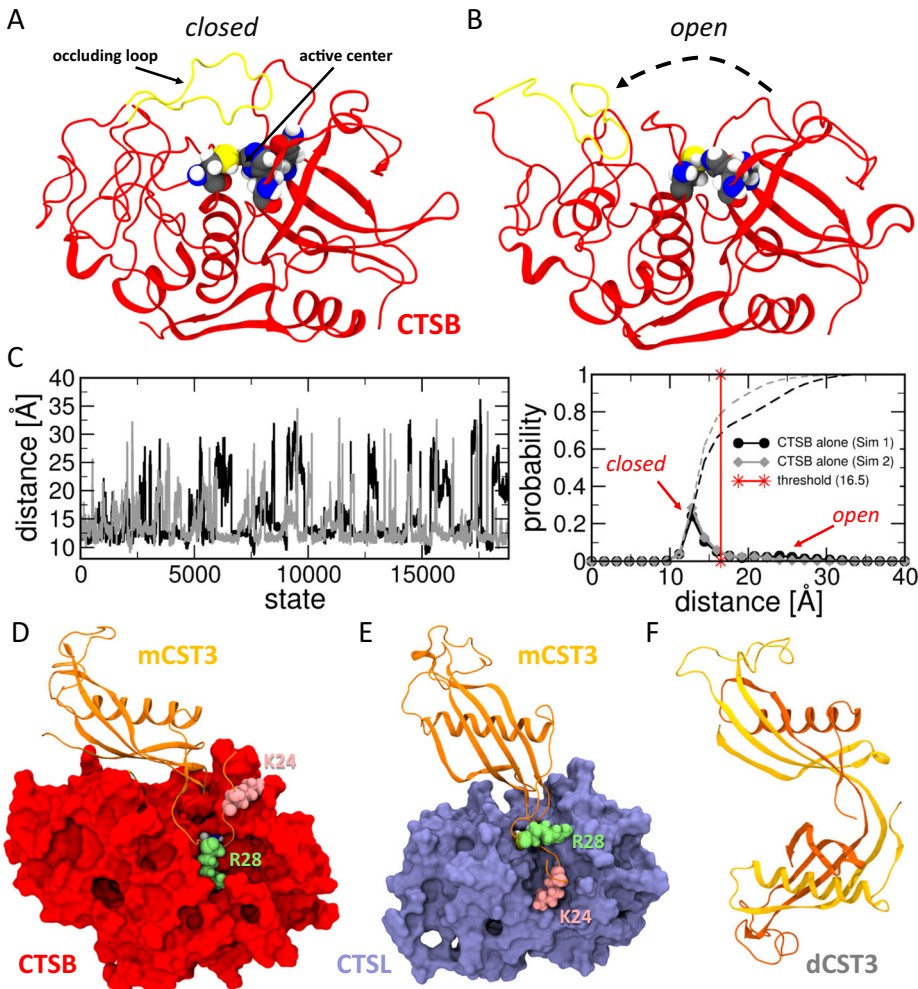

**Fig. 5 | Autoinhibition of isolated CTSB and inhibition complexes of CTSB and CTSL with mCST3 as obtained through molecular dynamic simulations.**
**A** Cartoon representation of murine cathepsin B structure (red) with the occluding loop (yellow) in a closed position. Active-site residues (C108, H278, N298) are shown as spheres and coloured by element (C: grey; N: blue; O: red; S: yellow; H: white). **B** Representative state of CTSB with an open conformation of the occluding loop. **C** (left) Centre-of-mass distance between the active site and occluding loop atoms across all structures generated during two independent TIGER2h$_{PE}$ simulations of isolated CTSB. (right) Histogram of distance data with the selected

threshold for open states (red). Dashed lines denote the integral and allow access to the open/closed fractions at different distance thresholds. **D** Inhibition complex of CTSB (red) with monomeric cystatin C (orange) as obtained through TIGER2h$_{PE}$ simulations. Residues K24 and R28, which are cleaved off by trypsin, are highlighted. **E** Inhibition complex of cathepsin L with cystatin C as obtained through TIGER2h$_{PE}$ simulations. **F** Cartoon representation of the cystatin C dimer structure (orange and yellow) illustrating the swap of the helical domain. The primary binding loop is now extended to connect the two monomers, therefore inhibition of CTSB is no longer possible. Source data are provided as Source Data file.

loop of CTSB was significantly increased when CST3 was bound to this allosteric site compared to the uncomplexed CTSB ($p = 0.0012$). As further support, the fraction of open states was increased by an average factor of 3.15 (Fig. 6B). Allosteric site prediction by the AlloSitePro algorithm for proteins resulted in the prediction of two sites, the active site and the binding site we identified by the MD-simulations (Fig. 6C)[25]. To verify that we have identified an unknown allosteric pocket at CTSB's surface, we carefully analysed the intermolecular interactions in these complexes with CST3 (Supplementary Figs. 8–11) and selected six of the most important residues of the proposed allosteric pocket for substitution (S169A, Y215F, E221V, D222P, H224L and D333P) (Supplementary Fig. 12b). Activity tests confirmed that the enzymatic activity of the CTSB mutant bearing six substitutions showed no enhanced activity in the presence of dCST3 (Fig. 6D), strongly indicating that the binding of dCST3 to this allosteric pocket in wild-type CTSB enhanced CTSB's activity and shifts the conformational equilibrium of the occluding loop towards an open state. This final cleavage construct dCST3-R28 still retained the N-terminal region that usually occludes the essential interacting residues of CST3 found

with the shorter constructs (R73 and K74) and we, hence, suspect unbinding of this N-terminal region is necessary to allow allosteric binding to CTSB. As it is assumed that pH shifts trigger the onset of AP, and as we demonstrated that the dimerisation of CST3 is dependent on both the pH and the truncation by trypsin (Fig. 4H, I), the release of the CST3 N-terminus must be favoured by these conditions, too. A schematic illustration summarises our findings and visualises the influence of CST3 and CST3 dimers on CTSB and CTSL activity in pancreatic acinar cells on trypsinogen activation in the secretory compartment (Fig. 6E).

**The disease severity of AP is increased in *Cst3*−/− mice**
Finally, we investigated the impact of CST3 on disease severity during experimental AP in *Cst3*−/− mice. AP was induced in *Cst3*−/− and wild-type mice by hourly injections of caerulein (50 µg/kg/bodyweight). H&E staining of pancreatic tissue sections showed that both mice strains developed AP within 8 h of caerulein treatment, but histologic scoring of 24 h sections detected an increased damage in the *Cst3*−/− mice (Fig. 7A), especially the number of necrotic cells and the number

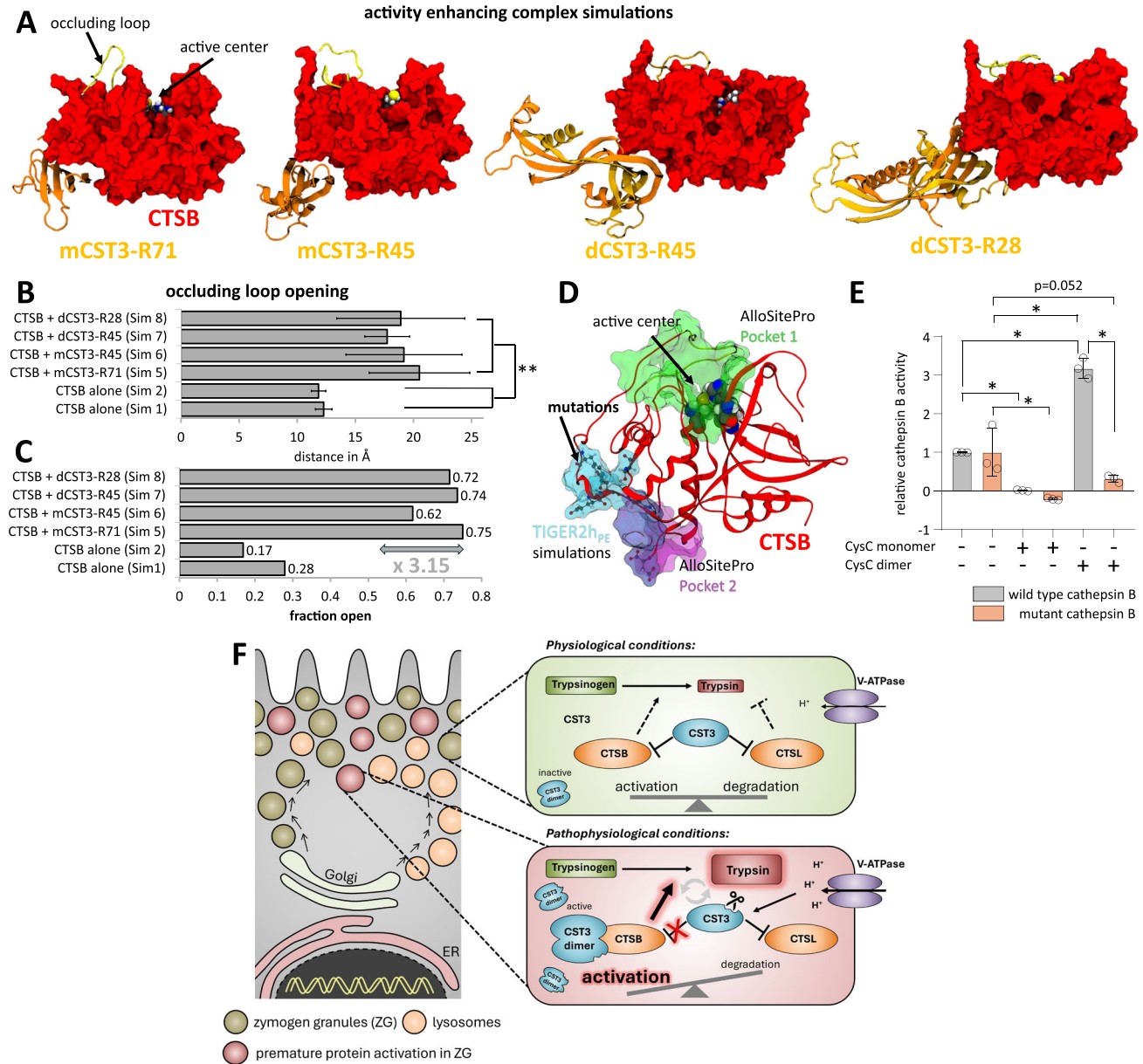

**Fig. 6 | Allosteric modulation of CTSB through dCST3 binding into the discovered pocket. A** Snapshot of major structural clusters as obtained by TIGER2h$_{PE}$ simulations of cathepsin B with different models of the trypsin-processed cystatin C. Binding to an allosteric pocket on CTSB's surface results in a pronounced opening of the occluding loop. **B** Average centre-of-mass distance (±SD) between the active site and occluding loop atoms for major cluster states generated throughout different TIGER2h$_{PE}$ simulations. Binding configurations into the allosteric site (four simulations) show significant differences to isolated CST3 (two simulations). Significance was calculated by one-way Anova followed by Dunn–Ši-dák correction for multiple testing ($p < 0.0001$ for all binding configurations into the allosteric site (sim5 $n = 5935$, sim6 $n = 8563$, sim7 $n = 980$, sim8 $n = 7026$) against isolated CST3 (sim1 $n = 7378$, sim2 $n = 5516$)). **C** Resulting fraction of open states using a distance threshold of 16.5 Å. Allosteric modulation by CST3 results in a 3.15-fold increase in active states. **D** Allosteric sites predicted by AlloSitePro (green, magenta) in superposition with the binding site predicted by TIGER2h$_{PE}$ simulations (cyan) and intersection surface area (purple). Mutation target selected based on simulation data are shown as ball-and-stick. **E** Activity tests with CTSB wild-type and mutated allosteric site (S169A, Y215F, E221V, D222P, H224L, D333P) under the influence of cystatin C monomer- and dimer-fractions. The wild-type shows a three-fold increase in activity when measured in complex with dimeric cystatin C, similar to the increase in open states. This activating effect is diminished with the allosteric site mutated (WT CTSB vs WT CTSB + CST3 monomer $p = 0.0048$, mut. CTSB vs mut. CTSB + CST3 monomer $p = 0.0009$, WT CTSB vs WT CTSB + CST3 dimer $p < 0.0001$, mut. CTSB vs WT CTSB + CST3 dimer $p < 0.0001$, WT CTSB + CST3 dimer vs mut. CTSB + CST3 dimer $p < 0.0001$). **F** Schematic illustration summarises the effect of CST3 processing for the balance of CTSB and CTSL activity and intracellular trypsinogen activation. Figure 6E shows three independent experiments; significance was calculated by one-way Anova followed by Dunn–Šidák correction for multiple testing. Results are shown as mean ± SD. Significance levels of $p < 0.05$ are marked by an asterisk. Source data are provided as Source Data file.

of infiltrating inflammatory cells were increased (Supplementary Fig. 13a–c). The disease severity marker serum amylase was higher 1 h and 24 h after induction of pancreatitis in *Cst3−/−* mice (Fig. 7B), and intrapancreatic trypsinogen activation was increased 1, 8 and 24 h after the onset of pancreatitis (Fig. 7C), while the activity of chymotrypsin was elevated at 1 h and 8 h (Fig. 7D). Increased intrapancreatic trypsin activity was associated with an elevated number of apoptotic acinar cells[11], shown by TUNEL staining, in the pancreas of *Cst3−/−* mice (Fig. 7E). In line with increased acinar cell damage, we could observe elevated levels of pro-inflammatory serum cytokines such as IL-6, IL-1β

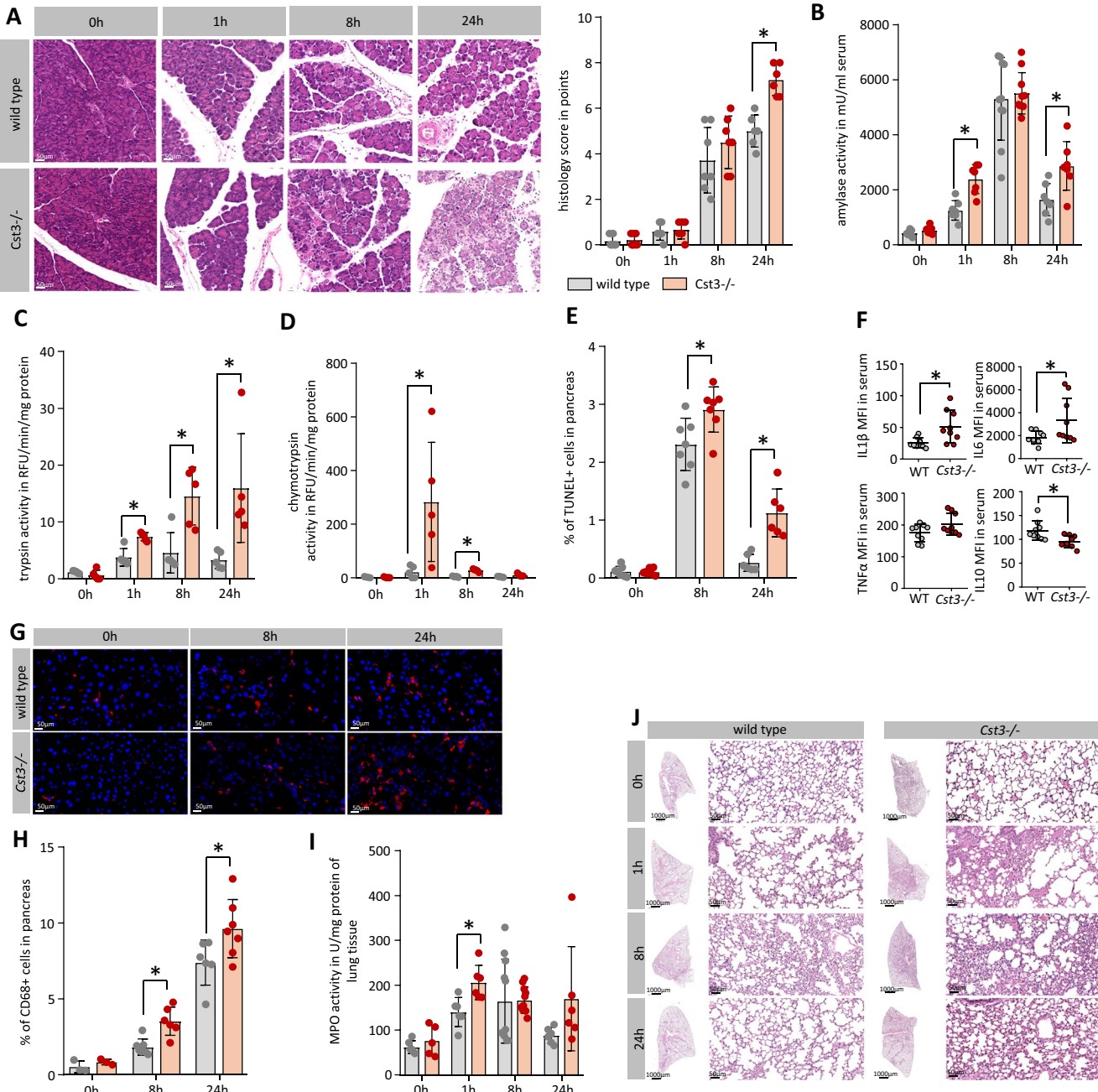

**Fig. 7 | The disease severity of AP is increased in Cst3−/− mice. A** H&E histology of pancreatic tissue illustrates the local damage in wild-type and *Cst3−/−* mice (scale bar represents 50 μm), histology score including oedema, necrosis and leucocyte infiltrating summarises the histological evaluation (0 h WT *n* = 6, *Cst3−/− n* = 7, 1 h WT *n* = 6, *Cst3−/− n* = 6, 8 h WT *n* = 7, *Cst3−/− n* = 7, 24 h WT *n* = 6, *Cst3−/− n* = 6) (24 h WT vs *Cst3−/− p* = 0.0002). **B** Measurement of amylase activity in serum of mice (0 h WT *n* = 8, *Cst3−/− n* = 8, 1 h WT *n* = 7, *Cst3−/− n* = 7, 8 h WT *n* = 10, *Cst3−/− n* = 10, 24 h WT *n* = 7, *Cst3−/− n* = 7) (1 h WT vs. *Cst3−/− p* = 0.0005, 24 h WT vs. *Cst3−/− p* = 0.0099). **C**, **D** Measurement of trypsin (1 h WT vs *Cst3−/− p* = 0.0014, 8 h WT vs *Cst3−/− p* = 0.007, 24 h WT vs. *Cst3−/− p* = 0.0193) and chymotrypsin (1 h WT vs *Cst3−/− p* = 0.0307, 8 h WT vs. *Cst3−/− p* < 0.0001) activity in pancreas tissue homogenate normalised to protein content (*n* = 5). **E** Quantification of apoptotic nuclei was performed from paraffin slides by counting TUNEL-positive nuclei in pancreatic tissue (0 h WT *n* = 7, *Cst3−/− n* = 7, 8 h WT *n* = 7, *Cst3−/− n* = 7, 24 h WT *n* = 6, *Cst3−/− n* = 6) (8 h WT vs. *Cst3−/− p* = 0.0206, 24 h WT vs. *Cst3−/− p* = 0.0007).

**F** Serum cytokines were measured 8 h after induction of pancreatitis by fluorescent beads. Dot blots show the mean fluorescent units (MFI) of the cytokines IL-1β (*p* = 0.0108), IL-6 (*p* = 0.0433), TNFα and IL-10 (*p* = 0.0087) (*n* = 9).
**G**, **H** Quantification of CD68+ macrophages was performed from cryo-embedded pancreatic tissue using immunofluorescent labelling (red, scale bar represents 50 μm) and illustrates the local immune response (0 h WT *n* = 3, *Cst3−/− n* = 3, 8 h WT *n* = 7, *Cst3−/− n* = 6, 24 h WT *n* = 6, *Cst3−/− n* = 7) (8 h WT vs. *Cst3−/− p* = 0.0017, 24 h WT vs. *Cst3−/− p* = 0.0408). **I** MPO activity in lung tissue homogenate (0 h WT *n* = 6, *Cst3−/− n* = 5, 1 h WT *n* = 6, *Cst3−/− n* = 6, 8 h WT *n* = 11, *Cst3−/− n* = 11, 24 h WT *n* = 6, *Cst3−/− n* = 6) (1 h WT vs *Cst3−/− p* = 0.0088) and (**J**) H&E staining illustrate increased lung inflammation in *Cst3−/−* mice (scale bar represents 50 μm an 1000 μm). **A**–**F** all data points represent biological replicates. Data represent five or more independent experiments; significance was calculated by two-tailed Student *t*-test for independent samples. Results are shown as mean ± SD. Significance levels of *p* < 0.05 are marked by an asterisk. Source data are provided as Source Data file.

and a decreased level of the anti-inflammatory cytokine IL-10 (Fig. 7F). A stronger inflammation, as observed by histology score and elevated serum cytokines, resulted in a higher number of intrapancreatic macrophages 8 h and 24 h after onset of disease in the *Cst3*−/− mice (Fig. 7G, H). In addition, we observed an elevated systemic inflammation, shown by increased myeloperoxidase (MPO) activity in lung tissue of *Cst3*−/− mice and affected lung histology (Fig. 7I, J).

### Genetic variants within the human CST3 gene

We analysed the presence and prevalence of *CST3* sequence variants in a cohort of pancreatitis patients and healthy controls. Due to a higher prevalence of genetic risk factors in patients suffering from recurrent or chronic disease, we analysed the potential role of *CST3* variants in CP patients. WGS-data from 51 idiopathic chronic pancreatitis (ICP) and 45 alcoholic chronic pancreatitis (ACP) patients were screened for promoter variants and missense mutations in the coding region of the *CST3* gene (Chr: 20 GRCh38: NC_000020.11 (23626706..23637955, complement)). The *CST3* gene is located in the cystatin locus and comprises 3 exons, spanning 4.3 kilo-base pairs. We identified the presence of a missense mutation (c.73C > T; p.A25T, rs1064039) a synonymous mutation c.12C > T; p.P4P and two sequence variants in the promoter region of the *CST3* gene, one of which (c.1-71A > C) was in almost perfect linkage disequilibrium with SNP rs1064039. We compared genotype data with available results from 408 individuals of the SHIP study, which is a population-based cohort study conducted in West Pomerania, the north-eastern part of Germany, and its details have been previously reported[26]. The SNP rs1064039 allele had a slightly increased prevalence in the CP cohorts in comparison to 408 SHIP controls (OR 1.1884 CI 0.8194–1.7236; *p* value 0.3628) (Supplementary Table 2), however, the association with CP was not statistically significant. It is known that the p.A25T variant influences the signal sequence cleavage[27]. For the SNP rs1064039 we performed specific TaqMan analysis in replication cohorts of 739 ICP cases and 695 blood donor controls (Supplementary Table 3). Also, in this larger cohort, a significant association of SNP rs1064039 with CP was not detected. These preliminary results could not confirm that genetic variants of the *CST3* gene play a role in the pathogenesis of chronic pancreatitis.

## Discussion

Acute pancreatitis has been characterised as the self-digestion of the pancreas by its own proteases. The activation of trypsinogen to trypsin is thought to play a central role in the onset of the disease[6,13]. The discovery of various mutations in the human *PRSS1* gene[8,28], or in genes involved in the processing or inhibition of trypsin activity such as *CTRC*[29] or *SPINK1*[30], confirm this hypothesis. Nevertheless, only a minority of patients suffering from acute or chronic pancreatitis carry a mutation in the trypsin activation/degradation pathway, while in the majority of cases, genetic factors could not be identified.

Intrapancreatic trypsinogen activation, in the absence of enterokinase, can be mediated by the lysosomal enzyme CTSB. Animal experiments provided evidence that the intracellular activation of trypsinogen is almost exclusively mediated by CTSB[6,7,11,12,31]. CTSL, another lysosomal cysteine protease, counteracts CTSB by degrading active trypsin and trypsinogen[17,31] and is believed to represent a cellular fail-safe mechanism against premature intracellular trypsinogen activation. In C57Bl/6 mice, the most frequently used mouse strain, an inborn functional defect in the *CTRC* gene[32] disrupts another important cellular protection mechanism of CTRC-dependent trypsin degradation, so the balance of the intracellular trypsin activity equilibrium is under the control of the two cysteine proteases CTSB and CTSL. Interestingly, under physiological conditions, both lysosomal enzymes are located in the secretory compartment of murine and also human acinar cells[11]. As a result, CTSB can be detected in the pancreatic fluid collection of patients[16]. Subcellular fractionation experiments clearly showed that CTSB and CTSL co-localise in the secretory

compartment of healthy mice without activating trypsinogen[11,33]. This finding is contradictory to the co-localisation hypothesis which claims that co-localisation with trypsinogen is a necessary pathophysiological requirement to induce protease activation. Two main theories have been proposed as to how the co-localisation of active CTSB and trypsinogen occurs, (1) by missorting of CTSB into the secretory pathway[15] and (2) via the fusion of zymogen granules with lysosomes[14]. Our findings suggest an alternative mechanism that is independent of missorting or fusion of vesicles. Subcellular fractionation experiments confirmed that under normal physiologic conditions, CTSB is already present in the zymogen-containing secretory compartment. CTSB and CTSL need no proteolytic activation by another protease, as they become autocatalytically activated during intracellular sorting[34]. However, the presence of the endogenous inhibitor CST3 prevents a CTSB-mediated activation of trypsinogen. In the presence of the inhibitor CST3, neither missorting, fusion of zymogen granules with lysosomes nor the physiological presence of CTSB in the secretory compartment would be sufficient to cause CTSB-mediated activation of trypsinogen. By necessity, the CST3-mediated inhibition of CTSB needs to be overcome for CTSB to activate trypsinogen during pancreatitis. Furthermore, the question arises, why CTSL does not counteract CTSB under these circumstances. Both cathepsins are inhibited by CST3 in the same manner. Subcellular fractionation experiments localised the activation of trypsinogen in the heavy fraction of zymogen granules[11,35]. It has also been shown that V-ATPase-mediated acidification of the zymogen vesicles is a necessary requirement for trypsinogen activation[19,36] and that acidic conditions reduce the inhibitory capacity of CST3 and promote dimer formation[20]. In our study we could show that the interaction of CTSB and CST3 is a pH-regulated process. A proteolytic processing of CST3 by active trypsin decreases its inhibitory capacity against CTSB, but not against CTSL. Surprisingly the activity of CTSB was even increased in the presence of processed CST3.

The cathepsins CTSB and CTSL are cysteine proteases and targets of type 2 cystatin protein inhibitors like CST3. However, the analysis of their protein structure revealed important differences. CTSB has a auto-inhibitory domain, the so called occluding loop, which can cover the active centre of the enzyme[23]. CTSB is known to have endo- and exopeptidase activity depending on the conformation of the occluding loop. If the occluding loop is in close vicinity to the active site and therefore CTSB forms a rather closed and compact structure, the exopeptidase activity is favoured, where CTSB removes dipeptides from the C-terminus of proteins or peptides. If the occluding loop moves away from the active site the endopeptidase activity is increased[37]. Our molecular analysis identified a second binding pocket of CTSB which can interact with CST3 dimers in its mature and processed form. This interaction stabilises the opened state of the occluding loop and therefore increases the proteolytic endopeptidase activity of CTSB. With a shift from monomeric to dimeric CST3 the balance changes from CTSB inhibition to overactivation. After tryptic cleavage, the binding of CST3 monomers to the active site becomes unstable and is overpowered by binding of the dimers to the allosteric pocket. At the same time, CTSL inhibition by the cleaved CST3 remains unchanged, due to the fact that the binding sites within the N-terminal domain are different compared to the binding sites with CTSB. Activation of CTSB and inhibition of CTSL appear to allow for efficient trypsinogen activation and the onset of pancreatitis.

Protease activation in wild-type and *Cst3*−/− animals occurred within minutes after caerulein treatment[11,38]. After disease onset *Cst3*−/− mice showed significantly more trypsin and chymotrypsin activities within the secretory compartment and had higher serum amylase activity in serum. CCK stimulation of freshly prepared *Cst3*−/− acinar cells also showed a higher protease activation. These results suggest that CST3 plays a critical role in the onset of pancreatitis by regulating intracellular CTSB-mediated trypsinogen activation. In patients,

elevated urinary TAP levels correlate with disease severity[39], and a number of studies, if not all, have found trypsinogen activation to correlate with disease severity in experimental pancreatitis. Trypsinogen T7 deficient mice showed a reduced disease severity compared to the studies of Halangk et al. in *Ctsb−/−* mice[7,40], whereas Geisz et al. and Chen et al. did not observe significant differences[12,31]. Interestingly in *Ctsl−/−* mice a higher trypsin activity was associated with a reduced disease severity[17], which could indicate that not trypsin activity, but cathepsin activity defines the severity, as it is known that especially the cytosolic activity of CTSB is able to induce apoptotic and necrotic cell death[11,41]. Notably, *Cst3−/−* mice did not develop spontaneous forms of acute or chronic pancreatitis, even in the presence of higher CTSB activity within the secretory compartment. One reason for this is probably the significantly increased and counteracting activity of CTSL[17] and the missing increased CTSB activity mediated by CST3 dimers. Furthermore, cathepsin activity is also regulated by acidification of the cellular compartment where trypsinogen activation takes place[19,36]. Without V-ATPase mediated acidification the activity of cathepsin does not reach a pathophysiological relevant level. Beside pancreatic acinar cells, also macrophage mediated trypsinogen activation is relevant during AP[42], the lack of CST3 in macrophages could also result in a higher CTSB mediated trypsin activity in infiltrating macrophages which increases the proinflammatory response, like we have observed, and can contribute to disease severity[3,5,42,43].

In humans CTSB, CTSL and CST3 have high protein sequence homologies of 79.5%, 72.1% and 69.2% with the murine counterparts. In our study of human pancreatic samples, we detected an analogous localisation of CST3 within the secretory compartment of acinar cells, which suggests a potential disease-relevant role of CST3 also in patients. Murine and human CTSB have nearly the same structural and biochemical properties[44]. Nevertheless, in our sequencing analysis of 739 pancreatitis patients, we could not identify genetic variations in the *CST3* gene that were significantly associated with chronic pancreatitis. Mutations that affect CST3 dimerisation like the L68Q variant[21] are, according to open databases, very rare and could not be found in CP patients. Larger cohort studies could clarify whether genetic variants in the *CST3* gene are associated with the occurrence of pancreatitis.

In conclusion, we could demonstrate that CST3, an endogenous inhibitor of CTSB and CTSL, regulates cathepsin activity in the secretory compartment of acinar cells. Our data emphasise that intracellular protease activation can occur independent of an active missorting of CTSB into the secretory compartment or an active fusion of zymogen-containing granules with lysosomes but via an alternative mechanism reported here. CTSB is physiologically localised in the secretory compartment, where its activity is controlled by CST3. It is the acidification of the secretory compartment which causes a diminished inhibitory capacity and dimer formation of CST3, that then triggers the activation of trypsinogen by CTSB. Furthermore, the tryptic cleavage of CST3 reduces the inhibitory capacity of monomers towards CTSB but not towards CTSL. CST3, the inhibitor of CTSB turns into its activator (dimeric CST3) and thereby shifts the balance towards trypsinogen activation. These early intracellular biochemical events represent a pathophysiological mechanism that can explain the initial intracellular protease activation by CTSB, independently of lysosomal fusion and missorting. Whether this activation mechanism has clinical implications for patients suffering from pancreatitis or even raises therapeutic options needs further evaluation.

## Methods

### Ethics statement
All animal experiments were previously approved by the local Animal Welfare Commission (LALLF - Landesamt für Landwirtschaft, Lebensmittelsicherheit und Fischerei Mecklenburg-Vorpommern, ref. nr. 7221.3-1-056/19) and performed in compliance with the ARRIVE guidelines. Studies on human samples were approved by Ethical Committee of the University Medicine Greifswald for SHIP (BB 39/08 and BB 122/13), GPCN and ProZyt (Reg.-Nr: III UV 05/06). The ChroPac trail (ISRCTN38973832) was approved by an independent ethics committee of the University of Heidelberg. All participants gave written informed consent. Participants were not compensated for their participation in the study.

### Animal experiments
C57Bl/6 mice were obtained from Charles River Laboratories (Sulzfeld, Germany), *Cst3−/−* mice (with a genetic C57Bl/6 background) were obtained from Efrat Levy, Ph.D. (Departments of Psychiatry, Biochemistry & Molecular Pharmacology, and the Neuroscience Institute NYU Langone Medical Center, Center for Dementia Research, Nathan S. Kline Institute for Psychiatric Research 140 Old Orangeburg Road, Orangeburg, NY 10962). All mice were kept under pathogen-free conditions in ventilated animal cabinets, with a 12-h light-dark cycle at a temperature of 21–24 °C (humidity 50–70%) with access to food and water ad libitum. Male and female mice with an age between 8 and 16 weeks were used for the experimental disease models. AP was induced by hourly i.p. injections of caerulein (50 µg/kg/bodyweight) up to a maximum of 8 h. Mice were sacrificed 1, 8 and 24 h after the first injection.

### Antibodies and reagents
The following antibodies were used for Western blot and immunofluorescent labelling: Anti-LIMP-1 (dilution 1:1000 for western blot, NBP2-67414, Novus Biological CO, USA), Anti-cystatin C (dilution 1:1000 for western blot, 1:100 for fluorescence labelling, ABC20, Merck Millipore, Burlington, MA, USA), Anti-syncollin (dilution 1:1000 for western blot, abcam ab178415, Cambridge MA, USA), anti-cathepsin B (dilution 1:1000 for western blot, 1:100 for fluorescence labelling, MAB965, R&D Systems MN, USA), anti-cathepsin L (dilution 1:1000 for western blot, MA5-23891, Thermo-Fisher Scientific, Bremen, Germany), anti-GAPDH (dilution 1:1000 for western blot, H86504M, Meridian OH, USA), anti-α-amylase (dilution 1:1000 for western blot, 1:200 for fluorescence labelling, SG46657, Santa Cruz TX, USA), anti-CD68 (dilution 1:100 for fluorescence labelling, ABIN181836, antibody online), anti-His-tag (dilution 1:1000 for western blot, ab18184, abcam, Cambridge, UK), pAb Rabbit anti-Mouse IgG (dilution 1:20000 Anogen, Toronto, Canada). The following regents were used: caerulein (C9026-1MG), and cholecystokinin fragment 26–33 amide (C2901-1MG; both from Sigma, Munich, Germany).

### Substrates
The cathepsin B activity was measured using Z-Arg-Arg-AMC (4004789.025, Bachem, Bubendorf, Switzerland) as substrate, cathepsin L activity was measured using Cbz-Phe-Arg-R110 (Invitrogen, Carlsbad, CA, USA) or Z-Phe-Arg-AMC (Bachem) as substrates, chymotrypsin using Suc-Ala-Ala-Pro-Phe-AMC (Bachem), and trypsin using R110-N-Cbz-Ile-Pro-Arg (Biotium, Fremont CA, USA). Protein concentration was determined by Bradford assay (SLCN3585 Sigma Aldrich-Chemie GmbH, Taufkirchen, Germany).

### Isolation of pancreatic acini
Pancreatic acini were isolated from murine pancreas by collagenase digestion 0.08 mg/ml (Collagenase of *Clostridium histolyticum* (EC.3.4.24.3) from Serva, lot no. 14007, Heidelberg, Germany)[11]. Cells were maintained and stimulated with 0.001 mM Cholecystokinin (CCK), CCK-Octapeptid (J66669.EXD Thermo-Fisher Scientific, Bremen, Germany) in Dulbecco's modified Eagle's medium containing 10 mM HEPES and 2% BSA. Activity of proteases was measured in living acini over a time period of 60 min as well as in acini homogenates after 30 min of stimulation with 0.001 mM CCK. Buffer for protease activity measurement in living acini contained 24.5 mM HEPES, 96 mM NaCl,

11.5 mM glucose, 6 mM KCl, 1 mM MgCl$_2$*6H$_2$O, 0.5 mM CaCl$_2$*2H$_2$O, 2.5 mM NaH$_2$PO$_4$*H$_2$O, 5 mM sodium fumarate, 5 mM sodium glutamate, 5 mM sodium pyruvate, and 1% BSA and DMEM at pH 7.4.

## Subcellular fractionation of the pancreas
Subcellular fractions were collected from pancreatic tissue of caerulein-treated mice and untreated animal controls. Fractionation was performed directly after sacrificing the mice[11,45]. The pancreas was cut into small pieces in ice-cold homogenisation buffer containing 240 mM sucrose, 5 mM MOPS, and 1 mM MgSO$_4$ (pH 6.5). The cellular fractionation was performed directly after cutting by two strokes of Douncer size A followed by two strokes of Douncer size B. The fractions were separated by density centrifugation at 4 °C according to the following steps. The post-nuclear supernatant was separated by $150 \times g$ centrifugation for 10 min. The zymogen granule-enriched fraction was centrifuged at $470 \times g$ for 15 min at 4 °C, followed by a lysosome-enriched fraction at $12,200 \times g$ for 12 min at 4 °C. The remaining supernatant was cleaned up by centrifugation at $20,800 \times g$ for 10 min and was used as the cytoplasmic fraction.

## Enzyme activity measurements in tissue homogenates
The activity of proteases was determined in acinar cell homogenate, subcellular fractions as well as total lysate of pancreatic tissue. Trypsin and chymotrypsin activity were measured as kinetic (60 min at 37 °C) in buffer containing 100 mM Tris and 5 mM CaCl$_2$ (pH 8.0) by a fluorometric assay using fluorochrome substrates. The trypsinogen and chymotrypsinogen content were determined in pancreatic homogenates after 30 min at 37 °C in the presence of 0.1 U/ml enterokinase (Sigma Aldricht-Chemie GmbH, Taufkirchen, Germany) by measuring trypsin and chymotrypsin activity. CTSB activity was measured in 100 mM sodium acetate and 5 mM CaCl$_2$ (pH 5.5) containing 10 mM DTT and at pH 4.0 for CTSL activity. Cathepsin activities were measured as kinetic over 1 h at 37 °C by a fluorometric assay using the specific fluorochrome substrates mentioned above. CTSB as well as CTSL activity under different pH conditions was measured in 100 mM sodium acetate containing 5 mM CaCl$_2$ and 10 mM DTT in a range of pH 3–8 with excitation at 340 nm and emission at 460 nm.

## Digestion of cystatin C by trypsin and chymotrypsin
Twenty microliter of a 0.5 mg/mL cystatin C solution was added to 227.5 μL trypsin buffer (100 mM Tris, 5 mM CaCl$_2$, pH 8.0) to a final cystatin C concentration of 40 μg/mL. 2.5 μL of a trypsin solution (Trypsin MS Grade, Promega, Wisconsin, United States) or chymotrypsin (α-Chymotrypsinogen A from bovine pancreas, Sigma Aldrich-Chemie GmbH, Taufkirchen, Germany) was added (final concentration of 0.1 μg/mL) and incubated for 1 h at 37 °C at 500 rpm. As a negative control, a cystatin C sample prepared the same way with a final concentration of 40 μg/mL was also incubated at 37 °C without the addition of trypsin. Instead, 2.5 μL of trypsin buffer was added. To identify the monomer/dimer-ratio of trypsin-mediated digested cystatin C, 14 mL of a 2 mg/mL cystatin C solution was rebuffered in trypsin buffer. The reaction was started by the addition of 60 μl of a 25 μg/mL trypsin solution and lasted 13 h at 37 °C at 180 rpm. Full cystatin C truncation by trypsin was verified by SDS-PAGE analysis.

## Activity tests of cathepsin B and cathepsin L in vitro
For activity tests, 1 or 2 μg/mL enzyme solutions of CTSB and CTSL were prepared. Substrate solutions with final concentrations of 400 μM Arg-Arg-AMC for CTSB and 30 μM Phe-Arg-AMC for CTSL activity supplemented with 10 mM DTT were prepared in CTSB or CTSL measuring buffer (100 mM sodium acetate, 5 mM CaCl$_2$, pH 5.5 or pH 4.0, respectively). Ten microliters of the enzyme solutions were mixed with either 10 μL of a solution of cystatin C monomer, cystatin C dimer, cystatin C monomer digested with trypsin, cystatin C dimer digested with trypsin, only trypsin or trypsin buffer. Reactions were

started by the addition of 90 μL of substrate solution and fluorescence was measured for 1 h at 37 °C. To have a better signal-to-noise ratio excitation occurred at 360 nm and emission at 470 nm.

## Determination of myeloperoxidase-activity in lung
MPO activity was measured in lung tissue homogenate. Lung tissue was homogenised in 20 mM potassium phosphate buffer (pH 7.4) on ice and centrifuged at 4 °C. The pellet, containing the neutrophils, was resuspended in ice-cold potassium phosphate buffer (50 mM, pH 6.0) containing 0.5% cetyltrimethylammoniumbromide. The suspension was frozen/thawed in 4 cycles, sonicated, and centrifuged at $20,000 \times g$. MPO activity was measured in 50 mM potassium phosphate buffer (pH 6) containing 0.53 mM O-dianisidine and 0.15 mM H$_2$O$_2$. The MPO activity was measured as kinetic over 10 min using a Spectramax spectrophotometer (Molecular Devices, San Jose CA, USA). MPO activity was calculated against protein content.

## Determination of serum amylase activity
Serum amylase activity was determined by using the colorimetric kit Amyl (Ref 11876473316) from Roche/Hitachi (Vienna, Austria).

## Apoptosis assay
The TUNEL assay was performed from paraffin-embedded murine pancreatic tissue by FragEL DNA Fragmentation Detection Kit (QIA39-1EA, Millipore/Calbiochem Billerica, Massachusetts, USA).

## Serum cytokine measurements
The serum cytokine concentration was measured by the LEGENDplex™ Multi-Analyte Flow Assay Kit (BioLegend, San Diego, CA, USA).

## Histology
H&E staining was done from paraffin-embedded pancreas and lung samples using 2 μm microtome sections. Immunofluorescence staining was performed on 2 μm cryo slides, fixed with acetone. Blocking was done by 20% FCS, primary antibody incubation was done in a 1:200 dilution in blocking buffer overnight at 4 °C. Secondary antibodies were also used in a dilution of 1:200 for 1 h. Cell nuclei were stained with DAPI. Tissue sections were scanned with a Pannoramic MIDI II (Sysmex) and analysed by Quant-Center-Software (Sysmex)[46].

## Genotyping of patients and blood donors
SHIP is part of the Community Medicine Research net (http://www.medizin.uni-greifswald.de/icm) of the University Medicine Greifswald, which is supported by the German Federal State of Mecklenburg–West Pomerania. The data set of the SHIP-2/Trend-0 cohorts used and analysed during the present study cannot be made publicly available owing to the informed consent of the study participants, but it can be accessed through a data application form available at https://fvcm.med.uni-greifswald.de/ for researchers who meet the criteria for access to confidential data.

The replication cohort consisted of blood donors (18–68 years, mean age 32.4 years, 58.2% male) who were recruited in 2010 at the University Medicine Greifswald. Written consent and information on alcohol and tobacco consumption, as well as a history of pancreatic disease, were recorded. Subjects with a self-reported history of pancreatitis were excluded from the analysis.

Replication analysis included also 739 German pancreatitis patients (5–97 years, mean age 46.6 years, 62.3% male) that had been enroled in an observational study (ProZyt) between 2003 and 2023. All individuals gave their written informed consent for participation and genotyping, and the local ethics committee approved the study protocol. Diagnosis of CP included the presence of unequivocal morphological evidence on computed tomography (CT) and/or endoscopic retrograde cholangial pancreatography (ERCP) and was labelled idiopathic chronic pancreatitis (ICP) in subjects in whom no known risk

factors, including gallstones, hyperlipidaemia or hypercalcaemia could be identified. Alcoholic aetiology (ACP: alcoholic chronic pancreatitis) was assumed in subjects with a history of regular alcohol consumption (more than two drinks or 20 g/day).

The cohort composition does not involve any selection bias and represents the local patient collective of patients suffering from chronic pancreatitis. Blood donors have been recruited during their visit in the Hospitals blood donation centre. Subjects with a self-reported history of pancreatitis were excluded from the analysis. The blood donor cohort was matched with the patient cohort concerning gender (~60% males). Different mean ages of patient and blood donor control cohorts were not expected to influence the prevalence of inherited genetic risk factors in both cohorts.

Genotyping was performed using TaqMan assay (Applied Biosystems) for rs1064039 and TaqPath Pro Amp master mix in a volume of 5 µL on a Qstudio7 Flex (Applied Biosystems). SNPs were in Hardy–Weinberg equilibrium ($p$ value > 0.001) and had a MAF > 0.01.

### Human pancreatic juice samples

Pancreatic fluid was collected within the GPCN study (Reg.-Nr: III UV 05/06) during or after interventional or surgical treatment of patients with pancreatic disease, the samples were collected during standard treatment of patients and not in context of a clinical trial. All individuals gave their written informed consent for participation and the local ethics committee approved the study protocol. The 18 samples include 11 patients with chronic pancreatitis, 6 patients with pancreatic cancer and 1 patient suffering from acute pancreatitis.

### Human chronic pancreatitis tissue samples

Human chronic pancreatitis tissue samples were collected in the context of the ChroPac trial (ISRCTN38973832)[47]. The study has been finalised and the results have been published[48].

### SDS-PAGE

Twelve microliter of the enzyme solution were mixed with 4 µL 4× loading buffer and the samples were denatured at 95 °C for 10 min. Subsequently, 15 µL samples and 8 µL Roti®-Mark Standard were loaded onto the gel. Either tris/glycine gels with 12.5% acrylamide resolving and 4% acrylamide stacking gel or tris/tricine gels with 10% acrylamide resolving and 4% acrylamide stacking gel were used.

### Western blot

An SDS-PAGE gel containing samples of digested and not digested cystatin C was blotted on a NC2 nitrocellulose membrane (0.2 µm, SERVA Electrophoresis GmbH, Heidelberg, Germany) via semi-dry transfer (150 mA, 30 min) using transfer buffer (25 mM Tris, 150 mM Glycine, 10% (v/v) methanol). A prestained protein ladder was used (PageRuler™ Prestained Protein Ladder, Thermo-Fisher Scientific, Waltham, USA). After blotting, the remaining binding surface was blocked using 5% (w/v) skimmed milk powder in PBS-T buffer (140 mM NaCl, 2.7 mM KCl, 10.3 mM $Na_2PO_4$, 1.8 mM $KH_2PO_4$ pH 7.4–7.6, 0.1% Tween-20) for 1 h at room temperature. Afterward, the NC2 nitrocellulose membrane was incubated with a 5% (w/v) skimmed milk powder solution in PBS-T buffer containing a 1:1000 dilution of the primary antibody. For this, the Anti-6X His-tag antibody was used and incubation occurred overnight at 4 °C. After three times washing for 5 min with PBS-T buffer the nitrocellulose membrane was incubated with the secondary antibody pAb Rabbit anti-Mouse IgG with a 1:5000 dilution in 5% (w/v) skimmed milk PBS-T buffer for 1.5 h. In the following, the membrane was washed again three times with PBS buffer before detection was done by using enhanced chemiluminescence (Carl Roth GmbH & Co. KG, Karlsruhe, Germany; OctopulsQPLEX, iNTAS science Imaging Instruments GmbH, Göttingen, Germany). As loading control Ponceau S stained lane intensity was used (VWR Chemicals, Solon, USA).

Pancreatic tissue or pancreatic fluid for Western blot analysis were lysed in buffer (25 mM HEPES, 75 mM NaCl, 0.5% Triton X-100, 5% glycerine, 1 mM EDTA) in the presence of 1 mM PMSF, 5 mM $Na_4P_2O_7$, 10 mM NaF, and 1 µg/mL aprotinin. Protein concentration was determined by Bradford protein assay. 50 µg protein was loaded on a polyacrylamide gel and transferred to a nitrocellulose membrane. Blocking was achieved in NET (150 mM NaCl, 5 mM EDTA, 50 mM Tris-HCl, and 0.05% Triton X-100) containing 2% gelatine. Detection of bands was performed with the Fusion-FXsystem (Vilber, Collégien, France) for chemiluminescence detection using ECL substrate (Thermo-Fisher Scientific).

### Mutagenesis of cystatin C

All cystatin C mutants were created by site-directed mutagenesis using the Q5® Site-Directed Mutagenesis Kit (New England Biolabs GmbH, Ipswich, USA). Non-overlapping primers were designed using the online tool NEBaseChanger and ordered from Invitrogen (by Thermo-Fisher Scientific, Waltham, MA, USA). The annealing temperatures suggested by NEBaseChanger were used for polymerase chain reaction (PCR). PCRs were performed according to the manufacturer´s protocol. The following primers and annealing temperatures were used for mutagenesis: CST3_K24A_fw 5′-CGCCACCCCGgcgCAGGGTCCGC-3′ ($T_A$ 66 °C); CST3_K24A_rv 5′-CTCTGGAAGTACAGATTTTCGCTG-3′ ($T_A$ 66 °C); CST3_R28A_fw 5′-ACAGGGTCCGgcgATGCTGGGTGCACCGG-3′ ($T_A$ 71 °C); CST3_R28A_rv 5′-TTCGGGGTGGCGCTCTGG-3′ ($T_A$ 71 °C); CST3_R44A_R45A_fw 5′-AGAAGGCGTGgcggcgGCACTGGACTT-3′ ($T_A$ 64 °C); CST3_R44A_R45A_rv 5′-TCATTGGCATCGGCTTCT-3′ ($T_A$ 64 °C); CST3_K24A_R28A_fw 5′-GCAGGGTCCGgcgATGCTGGGTGCACCGGA AGAAGC-3´ ($T_A$ 72 °C); CST3_K24A_R28A_rv 5′- CCGGGGTGGCGCTC TGG-3′ ($T_A$ 72 °C).

First, the cystatin C mutants K24A, R28A, and R44A_R45A were created using one primer pair. Further combinatorial mutagenesis of the four residues K24, R28, R44, and R45 was performed by combining a plasmid that already contained one substitution with a convenient primer, e.g. for the creation of the mutant K24A_R44A_R45A the plasmid of K24A was used as a template and PCR was performed with the primers for the mutagenesis of R44A_R45A. For the creation of the mutant where all four possible cleavage sites were substituted the plasmid of K24A_R44A_R45A was used as a template and PCR was performed with the primer pair containing the nucleotide sequence for K25A and R29A.

The generated constructs were transformed in *E. coli* Top10 (Thermo-Fisher Scientific, Waltham, MA, USA) using the heat-shock method. The next day, several colonies were picked and used to inoculate 4 mL LB media supplemented with 50 µg/mL kanamycin. The plasmids of the cultures were isolated on the next day according to the protocol of innuPREP Plasmid Mini Kits (Analytik Jena, Jena, Germany). To verify the desired plasmid sequence, the plasmids were sequenced using Mix2Seq Kits (Eurofins Genomics, Ebersberg, Germany). Plasmids containing the desired construct were transformed into *E.coli* SHuffle T7 Express (New England Biolabs GmbH, Ipswich, USA) for recombinant expression as described above. The expression yields are listed in Supplementary Tables 4 and 5.

### Protein expression and purification

Synthetic genes for the lysosomal proteases cathepsin B and L and the inhibitor cystatin C from mouse were ordered in a pET28a(+) vector (BioCat GmbH, Heidelberg, Germany) and verified by sequencing. All genes contained an N-terminal His-tag sequence for affinity chromatography and a TEV protease cleavage site to have the possibility to remove the His-tag. The sequences and NCBI accession codes are listed in Supplementary Table 6. The sequences for the proteases cathepsin B and L also contained the propeptides to prevent unwanted protein activity during the expression but no signal peptide. The sequence for cystatin C was also ordered without the signal peptide. The protein

expression and purification were performed according to a previously published protocol[49].

For protein expression, the pET28a(+) vectors harbouring the synthetic genes were transformed into *E. coli* SHuffle® T7 Express by the heat-shock method. For each gene expression, a single colony of the cells with the desired amino acid sequence was picked and used to inoculate 4 mL TB media that was supplemented with 50 μg/mL kanamycin. The overnight cultures were grown at 30 °C at 140 rpm. These cultures were used the next day to inoculate 200 mL main cultures in which the cells were grown until an optical density at 600 nm (OD600) reached approximately 0.6–0.8. Afterwards, the protein expression was induced with a final concentration of 0.1 mM isopropyl-β-D-thiogalactopyranoside (IPTG) for 24 h at 16 °C at 140 rpm. The main cultures were harvested by centrifugation (10 min, 2000 × *g*, 4 °C) and washed once with sodium phosphate buffer (50 mM, pH 6.0). The harvested bacteria pellets were resuspended in 4 mL equilibration buffer (50 mM sodium phosphate, 300 mM NaCl, 10 mM imidazole, pH 8.0) for each gram of cell pellet. The cells were disrupted via ultrasonication with 30% power and 50% cycle on ice. Thereby, the sonication procedure consisted of 5 min sonication followed by a 2 min break and another 5 min of sonication. Afterwards, the samples were centrifuged at 10,000 g for 40 min at 4 °C for separation of the cell debris from the supernatant. The clarified lysates containing the desired proteins were transferred onto Ni-IDA columns for affinity chromatography, washed ten times with washing buffer (50 mM sodium phosphate, 300 mM NaCl, 20 mM imidazole, pH 8.0), and eluted in fractions with elution buffer (50 mM sodium phosphate, 300 mM NaCl, 250 mM imidazole, pH 8.0). The fractions with the highest protein content were pooled and rebuffered in CTSB buffer (100 mM sodium acetate, 5 mM calcium chloride, pH 5.5) using Amicon Ultra-15 centrifugal filter units (MWCO 10kD; Millipore, Burlington, Massachusetts, USA). The protein solutions were stored at 4 °C until further use.

### Size exclusion chromatography

Cystatin C monomer and dimer fractions were separated by size exclusion chromatography (SEC) using a Superdex 75 Increase 10/300 GL column installed on an ÄKTA pure™ chromatography system (Cytiva, Freiburg, Germany). As a running buffer, the CTSB buffer supplemented with NaCl (100 mM sodium acetate, 5 mM calcium chloride, 100 mM NaCl, pH 5.5) was used.

### Protease activation

Since cathepsin B and L were expressed as proenzymes, the propeptides have to be cleaved off to get mature and active protein. The N-terminal propeptides are shielding the active sites keeping the protein inactive. Both proteins are able to cleave their own propeptide and activate themselves. Therefore, after rebuffering in CTSB buffer (100 mM sodium acetate, 5 mM CaCl₂, pH 5.5), 0.5 mg/mL CTSB and CTSL solutions were prepared and 10 μL of a 1 M DTT solution were added (final concentration 10 mM). Afterwards, the protein solutions were incubated at 37 °C at 500 rpm overnight. The activation was analysed by SDS-PAGE.

### Protein yields

The protein concentrations were measured via NanoDrop 1000 (Thermo-Scientific, Wilmington, DE, USA), and protein yields were calculated based on the molecular weight and the extinction coefficients from the expasy tool ProtParam (https://web.expasy.org/protparam/).

### pH-dependent dimerisation of cystatin C and trypsin-cleaved cystatin C

After the purification step or trypsin cleavage, the same amounts of cystatin C or truncated cystatin C, originating from the same expression batch, were rebuffered in seven different buffers over a pH range from 4.0 to 8.0. The used buffers were sodium acetate buffers (100 mM sodium acetate, 150 mM sodium chloride, pH 4.0, 5.0 and 5.5, respectively) and sodium phosphate buffers (100 mM sodium phosphate, 150 mM sodium chloride, pH 5.5, 6.0, 7.0 and 8.0, respectively). The protein solutions were stored at 4 °C in the corresponding buffer overnight. On the next day, the ratio of monomer and dimer fractions of the 3 mg/mL protein solutions in the different buffers was analysed using size exclusion chromatography.

### Molecular dynamics simulations/model building and positioning

Structures for murine cathepsin B (80–333) and cystatin C (21–140) were generated with Alphafold2[50] and returned very similar to available human or bovine crystallographic structures (Supplementary Fig. 14). Residues were numbered according to Uniprot[51], accession codes P10605 and P21460, for murine CTSB and CST3, respectively. Initial inhibition complexes were built in analogy to papain and cystatin B by sequence alignment. A human cathepsin L (114–333) structure was taken from PDB 7QKC and the initial CST3 position was found through sequence alignment with 1STF, as described previously. A human cystatin C dimeric structure was taken from PDB 1R4C (37–146), Uniprot accession code P01034. For simulations of CTSB enhancement, the initial murine cystatin C cleavage construct (mCST3-R71) was positioned on the rear side of CTSB, opposite the active site. This placement was chosen to avoid bias toward regions near the active centre and promote broader sampling (Supplementary Table 1, Simulation 5). Subsequent simulations always started the CST3 fragment aligned with the formerly obtained pose, but an originally closed CTSB. This was to sample for a stable binding pose and check for reproducibility of the occluding loop opening. Convergence of conformational ensembles is demonstrated in the supporting information (Supplementary Fig. 15).

### Open/closed states of CTSB

The occlusion loop states were analysed based on the centre of mass (COM) distances between the occluding loop (residues 185–200) and the active site (residues 108, 278, 298), indicating whether the active site was covered based on distance thresholds obtained from the isolated CTSB.

### TIGER2h_PE replica-exchange simulations

Conformational and configurational state ensembles were collected by TIGER2h_PE simulations[22]. The CHARMM forcefield and the CHARMM-GUI[52,53] are used to parameterise and truncate the proteins, and to add disulfide bonds according to Uniprot. Hydrogen mass repartitioning (HMR) was applied with psfgen included in VMD 1.9.4[54] and the time step was increased to 4 fs. The systems were solvated with the TIP3P water model and neutralised with sodium or chloride ions using VMD. The cell dimensions were selected cubic and the side lengths are reported in Supplementary Table 1. All simulations were performed with NAMD 2.14[55] using its TCL programming language API and the Colvars module[56]. Full periodic boundary conditions are applied and long-range electrostatics are considered with PME in an pseudo-NPT ensemble[57]. The sampling and cooling phases were set 16 ps and 8 ps, respectively. The temperature ladder spanned from 300 K to 450 K. Pressures and temperatures were maintained by a Langevin piston barostat at 100 fs period and 200 fs decay time and a Langevin thermostat with a 1 ps⁻¹ damping coefficient. RMSD flat-bottom restraints were applied on the backbone atoms of the proteins, except the occlusion loop of CTSB (residues 185–200) to restrict the sampling local to the starting structure (Supplementary Table 1). Implicit solvent energies during exchange decisions were evaluated by the GB_OBCII model using OpenMM[58] and periodic boundary conditions, while non-bonded cutoffs were set to half the shortest cell dimensions. The

replica-exchange simulation started after 60,000 steps of energy minimisation.

## Filtering of structural ensembles by contacts

To improve subsequent cluster analysis results, structural ensembles of complexes obtained through TIGER2h$_{PE}$ were filtered by contacts between both proteins, discarding structures without a contact within 0.33 nm. To analyse binding into the allosteric pocket, states were additionally filtered for contacts between the respective cystatin C fragment and the allosteric pocket residues used for mutagenesis found from the construct of murine mCST3-R71 (Supplementary Table 1, Simulation 5).

## Statistical analysis of residue contacts

Contact statistics in complexes were evaluated by counting individual residue contacts between both proteins. A contact is characterised by two atoms coming closer than 0.33 nm. The counts are normalised to the number of frames and represent the average fraction of contacts per residue throughout the simulation. A value larger than one means one residue binds to multiple residues at a time. Such interaction statistics are visualised as averages over protein surfaces or displayed in contact networks to emphasise important residues and regions for the interaction[59].

## Contact-contact principal component analysis (ccPCA)

To identify favourable binding structures of cystatin C and its trypsin cleavage constructs to the surface of CTSB or CTSL, we utilised a contact-contact principal component analysis (ccPCA). An in-house python code loads the molecular systems and resulting structural ensembles from TIGER2h$_{PE}$ and selects a suitable contact mesh between both proteins. Based on the number of atoms (excluding hydrogen), the minimal number of particles from the smaller protein is then used with a corresponding stride value to select the same number of particles in the other protein and distances between all such pairs over time are collected. This data is subjected to PCA and sorted onto a 2D histogram from the 1$^{st}$ and 2nd principal components. The joint probability distribution is converted to the Gibbs free energy (Eq. 1) as:

$$\Delta G = -RT \cdot \ln\left(\frac{P}{\max(P)}\right) \qquad (1)$$

for each state, where R is the gas constant and P is the probability of a state in the PCA space. Subsequently, the data is scanned for clusters within the first two principal components using the density-based OPTICS method from scikit-learn[60].

## Allosteric Site prediction with AlloSitePro

Prediction of allosteric sites for CTSB to crosscheck TIGER2h$_{PE}$ results was conducted for the mature murine CTSB sequence, excluding all propeptides (80–333) using AlloSitePro[25].

## Reporting summary

Further information on research design is available in the Nature Portfolio Reporting Summary linked to this article.

## Data availability

The initial configuration. AlphaFold2 predictions and full conformational ensembles (trajectories) from MD simulations are provided at Zenodo. Uniprot accession codes referenced in this work are P10605 (murine CTSB), P21460 (murine CST3) and P01034 (human CST3). Reference structures used in this work are available in the PDB under accession codes 1STF (inhibition complex of human stefin B with papain), 7QKC (human cathepsin L) and 1R4C (dimeric truncated human cystatin C). The data set of the SHIP-2/Trend-0 cohorts used and analysed during the present study cannot be made publicly available owing to the informed consent of the study participants, but can be accessed through a data application form available at https://fvcm.med.uni-greifswald.de/ for researchers who meet the criteria for access to confidential data. The genotyping analysis of pancreatitis patients is an ongoing cooperative research project, so the full sequencing data cannot yet be made publicly available. Individual requests can be made to the corresponding author (Matthias.sendler@uni-greifswald.de). Source data are provided as a Source Data file. Source data are provided with this paper.

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

## Acknowledgements

The authors would like to thank Levy Efrat (Center for Dementia Research, Nathan S. Kline Institute for Psychiatric Research, Orangeburg, NY, 10962, USA) for providing the *Cst3*–/– mice. We also thank Martin Kulke for the fruitful discussion and Kathrin Gladrow, Diana

Krüger, Jenny Radel and Susanne Wiche for their technical support. This work was supported by: Deutsche Forschungsgemeinschaft DFG SE 2702/2-1 - M.S. Deutsche Forschungsgemeinschaft DFG SE 2702/2-3 - M.S. Deutsche Forschungsgemeinschaft GRK 2719 B7 - M.S. Deutsche Forschungsgemeinschaft GRK 2719 B7 - A.A.A. Deutsche Forschungsgemeinschaft GRK 2719 C3 - U.B. Deutsche Forschungsgemeinschaft GRK 2719 B1 - U.V. Deutsche Forschungsgemeinschaft GRK 2719 B3 - M.L. Deutsche Forschungsgemeinschaft GL 1096/1-1 - J.G. PePPP centre of excellence MV (ESF/14-BM-A55-0045/16) – M.M.L., F.U.W.

## Author contributions

Concept of the study M.S., F.U.W., J.M.M., C.M., S.A. and U.B. Data acquisition and interpretation: J.M.M., C.M., S.A., N.G., N.R., M.D., H.W., L.S., A.S., L.G., D.B., E.H., J.G., O.S., M.L., M.D.C., U.V., A.A.A., M.M.L., F.U.W., U.B. and M.S. Writing committee M.S., C.M., N.G., S.A., F.U.W. Correction of manuscript and approval of final version: all.

## Funding

## Competing interests

The authors declare no competing interests.
