## [Transparent Peer Review file · Nature Communications]

Biochemical Analyses of Cystatin-C Dimers and Cathepsin-B reveals a Trypsin-Driven Feedback Mechanism in Acute Pancreatitis

Corresponding Author: Dr Matthias Sandler

Version 0:

Reviewer comments:

Reviewer #1

(Remarks to the Author)

Modenbach et al. describe the dysregulation of cathepsin B (CTSB) by cystatin C (CST3). Increased activity of CTSB in pancreatic acinar cells can promote acute pancreatitis by activating trypsinogen in the zymogen granules of these cells. CST3, a small protein also present in the zymogen granules, inhibits CTSB by binding to its active site, thereby preventing the onset of pancreatitis. CST3 can exist as a monomer or dimer, depending on the pH. Dimerized CST3 cannot function as an inhibitor; instead, it stimulates CTSB activation by binding to an allosteric site. Trypsin can further degrade CST3, promoting CTSB activation. Dimerization or proteolysis of CST3 does not change its inhibitory activity against cathepsin L, which further increases trypsinogen activation. The manuscript is well-written and contributes significantly to the understanding of the molecular basis of pancreatitis.

Comments:

1. It is mentioned in the Introduction that CTSB can activate trypsinogen to trypsin in acinar cells, but it is not mentioned that trypsin is responsible for the activation of other zymogens including chymotrypsins. It is important because chymotrypsin activity was measured in the study.
2. It is mentioned in the results that CST3 undergoes redistribution into zymogen granules in AP patients, but this is not compellingly demonstrated in Figure 1H. The levels of CST3 in pancreatic juice samples are variable and do not correlate with the levels of CTSB. The results also suggest that trypsin degrades CST3. Trypsin activity measurements of pancreatic juice samples would help clarify this phenomenon.
3. Line 138: "we measured enzyme activation" instead of "we measured enzyme activities" would be more correct as enzyme activities over time are presented in Figure 2E.
4. Figures 2B and 2D show enzyme activities in units RFU/mg and RFU/ μ g. It seems that the time component is missing from the figures (RFU/min/mg and RFU/min/ μ g). It is not clear why the activity values for trypsin and chymotrypsin are largely different by comparing Figures 2B and 2D.
5. The use of chronic pancreatitis patients for the genetic studies should be clarified, given that CST3 deficiency is associated with acute pancreatitis.
6. Does CST3 contain any disulfide bonds? Disulfide bonds may prevent the degradation of the inhibitor, even if it is cleaved.
7. Protease activities were measured for 60 min at 37 °C. How the evaporation of the samples was prevented? Was the kinetic linear over the course of these long measurements?
8. Line 217: The abbreviation MDS is unnecessary as it appears only once in the text. Line 366: Sentence starting with "Trypsinogen T7 deficient mice..." is incomplete. Words are missing from the end of the sentence.

9. If independent samples were measured, using the mean \pm SD would be more appropriate than mean \pm SEM.

10. The manuscript contains some typos such as promoter (line 294), with (line 415).

Reviewer #2

(Remarks to the Author)

This manuscript investigates the role of cystatin C during acute pancreatitis and suggests a novel allosteric interaction with cathepsin B. Overall the results are interesting and important, but addressing the below queries would strengthen the conclusions.

-Figure 1B – Please provide the whole blot for cathepsin B, which includes the size of the pro-form, single and heavy chain. It would also be helpful to see a blot for cathepsin L, given differences in activity shown in 1D. What is the purpose of the dashed line in the gel? It gives the impression that the gel is cropped, but if this is not the case, suggest removing it.

-Line 119 and Figure 1H – the text states pancreatic juice from patients. Please specify what patients? Do they have pancreatitis? No information about these samples is provided in the methods section.

-In Figure 2F, is there a significant difference between cathepsin B activity in the ZG of WT mice between untreated and pancreatitis conditions? If so, this should be indicated. This same comparison is shown to be significant in 1C, but it's not clear if this is consistent given the spread of the data. In Fig 3A, there is no difference at pH 5.5 (the pH of 1C and 2F). Why is it inconsistent?

-If trypsin and chymotrypsin are driving acinar death, then why is there no difference in necrosis between WT and Cst3^{-/-}, where there is significantly more activation of these enzymes?

-What is the relationship between cathepsin B/L and chymotrypsin? There is a much bigger difference in chymotrypsin activation in the absence of cystatin C compared to trypsin. It would thus be critical to show whether chymotrypsin can cleave cystatin C.

-Fig 3G-H, please indicate the assay pH in the figure legend (this would be helpful to include throughout).

-Fig 3A-D Please indicate in the figure legend what the red line on the graphs represents. It is not clear – presumably the pH optimum for each enzyme? If so, it is in the wrong spot for cathepsin B (shown to be 5.5 in Fig S1).

-Also in Figure S1, why does cathepsin L have no activity at pH 6.5 but it's active at 7? If there were no data points for 6.5, this should be indicated to avoid confusion. Error bars for this graph would also be helpful.

-Given the R28 truncation occludes the essential cathepsin B-interacting residues, how does the trypsin-cleaved version (which is primarily cleaved at this site, along with K24) enhance cathepsin B activity? It would be important to show experimentally that the shorter truncations promote activation even further (e.g., R71, 73, 74).

-Does the truncation somehow promote dimerization? That seems like what the model is suggesting, but there is no evidence for this.

-What proportion of cystatin C is in monomer v dimer form in the pancreas? It would be important to show that this balance is skewed towards dimer after pancreatitis initiation. Moreover, experimental evidence that cystatin C is actually truncated in vivo during pancreatitis is lacking. If this is a physiologically relevant mechanism, would you not expect to see cleaved cystatin C in the western blot shown in 1B? There doesn't appear to be, or at least not as dramatic as observed in the cleavage assays. There may be some evidence of cleavage in the human pancreatic juice, but it's not clear that this is from patients with pancreatitis, and it is extra-granular so may not represent what is actually happening inside the pancreas. Proteomics/N-terminomics on ZG proteins with/without caerulein would reveal the proportion of cleaved/uncleaved cystatin C (if any).

-The discussion indicates that low pH induces cystatin C dimerization. The main differences in cathepsin B activity upon caerulein treatment were observed above pH 6.5. In that case, it would be important to look at dimerization of the cleaved cystatin 3 across a wide pH range.

-The potential allosteric site within cathepsin B is very interesting, but whether or not it is relevant to pancreatitis is still unclear. These data might be more compelling as two separate manuscripts, where the novelty of the interaction between cystatin c and cathepsins is the focus.

-The occluding loop of cathepsin B is important for governing its endo and exopeptidase activity (which is pH dependent). It would be worth discussing how cystatin C binding may shift the substrate repertoire.

-Cystatin C can also inhibit legumain through a distinct binding site. It would be important to mention this and discuss

potential implications.

-CSTB is often used instead of CTSB (CSTB is stefin B). CTS3 is used in place of CST3. Cys3^{-/-} is used in some of the figures instead of Cst3 (e.g. 3B,D). Please amend.

-Sometimes bar graphs show only error bars, while others show the mean, SEM, and the individual data points. The latter is much more useful, so it would be helpful to use that style consistently.

Reviewer #3

(Remarks to the Author)

In a study entitled „Biochemical Analyses of Cystatin-C Dimers and Cathepsin-B reveals a Trypsin-Driven Feedback Mechanism in Acute Pancreatitis“ the authors provide compelling evidence that co-localization of the cysteine protease CTSB with trypsinogen within the zymogen-containing secretory compartment is not sufficient for trypsinogen activation since that under physiological conditions CTSB activity is regulated by the type 2 cystatin protein inhibitor, CST3, making it a key enzyme in the development of pancreatitis. Using computational methods, the authors provided an understanding of the autoinhibitory feature of CTSB and the mechanism by which the CST3 monomer effectively supports this autoinhibition by preventing exposure of the active site. Furthermore, the dimeric form of CST3, together with tryptic degradation products, promotes conversion to trypsin by binding to the allosteric binding site and stabilizing the occluding loop in an open conformation, thus making the active site more receptive to trypsinogen binding.

Although the performed MD simulations and the way the authors presented the results reliably support the experimental observations, I have a few questions regarding the computational part:

- 1) Is there a particular reason why the modelling was done on murine rather than human proteins, especially if human crystal structures are available? What is the percentage of sequence identity between human and murine CTSB and CST3?
- 2) Did the authors generate the initial CTSB:CST3 complex using AF2 or did they model the proteins individually and then obtain the complex by aligning the structures with that of PDB 1STF?
- 3) Is there any particular reason why restraints were applied on the all backbone atoms (except the occlusion loop of CTSB)? Could this have an effect on the assessment of allosteric interactions?
- 4) When authors say that the occluding loop has shifted from its usual position upon binding of monomeric CST3, does this mean that the active site is still protected, but this time by the CST3 N-terminus?
- 5) Could the authors show the interaction network of CTSB and CST3 during inhibition for the dominant conformation (Cluster 2), instead of Cluster 4, since according to Fig. S3 (E) K24 appears to be quite important for CTSB binding, while R28 also established a contact for more than 25% of the simulation time? In comparison, in the CST3:CTSB simulation in Cluster 0, K24 does not form interactions that persistent more than 25% of the time (Fig. S2 (F)). What suggests that the N-terminal region of CST3 is not as important for CTSB binding compared to CTSB?
In Fig. S3 (E), CTSB should be written instead of CTSB.
- 6) It is not clear to me why CST3 fragments cleaved at position R71 and R45 were computationally studied if K24 and R28 were found to be tryptic cleavage sites?
- 7) How the initial complexes of the processed m-/h- monomeric/dimeric CST3 with CTSB were generated – by docking, by using AF2?
- 8) When human CST3 in dimer form was simulated, was it simulated in complex with human or murine CTSB (Fig. 6 (A))?
- 9) Have the authors tried to model tryptic CST3 fragments with CTSB as well? Do they bind in the allosteric site or the orthosteric site was retained?

Version 1:

Reviewer comments:

Reviewer #1

(Remarks to the Author)

The authors addresses all my previous concerns and incorporated the suggested corrections into the manuscript. The revised version reflects significant improvements in clarity, accuracy, and scientific rigor. I have no further comments, and I am satisfied with the current version of the manuscript.

Reviewer #2

(Remarks to the Author)

The reviewer thanks the authors for addition of extra experiments, which have strengthened the conclusions of the manuscript.

Reviewer #3

(Remarks to the Author)

The authors have addressed all questions in the resubmitted manuscript and response to reviewers. I have no additional comments.

Reviewer #1 (Remarks to the Author):

Modenbach et al. describe the dysregulation of cathepsin B (CTSB) by cystatin C (CST3). Increased activity of CTSB in pancreatic acinar cells can promote acute pancreatitis by activating trypsinogen in the zymogen granules of these cells. CST3, a small protein also present in the zymogen granules, inhibits CTSB by binding to its active site, thereby preventing the onset of pancreatitis. CST3 can exist as a monomer or dimer, depending on the pH. Dimerized CST3 cannot function as an inhibitor; instead, it stimulates CTSB activation by binding to an allosteric site. Trypsin can further degrade CST3, promoting CTSB activation. Dimerization or proteolysis of CST3 does not change its inhibitory activity against cathepsin L, which further increases trypsinogen activation. The manuscript is well-written and contributes significantly to the understanding of the molecular basis of pancreatitis.

Comments:

1. It is mentioned in the Introduction that CTSB can activate trypsinogen to trypsin in acinar cells, but it is not mentioned that trypsin is responsible for the activation of other zymogens including chymotrypsins. It is important because chymotrypsin activity was measured in the study.

Response: We added the following sentence into the introduction: “Trypsin, in turn, proteolytically activates the other zymogens, including chymotrypsinogens, proelastases, and procarboxypeptidases”.

2. It is mentioned in the results that CST3 undergoes redistribution into zymogen granules in AP patients, but this is not compellingly demonstrated in Figure 1H. The levels of CST3 in pancreatic juice samples are variable and do not correlate with the levels of CTSB. The results also suggest that trypsin degrades CST3. Trypsin activity measurements of pancreatic juice samples would help clarify this phenomenon.

Response: We analysed 18 additional samples of human pancreatic fluid secretions to determine whether trypsin activity in pancreatic fluid has an influence on the presence of CST3. These samples had been collected from drainage of the pancreatic duct after surgical procedures on the pancreas, and were taken over days in a non-standardized procedure, which may have influenced the activation status of pancreatic zymogens. The activities of trypsin and cathepsin B in these samples were measured and correlated with Western-blot data of cystatin C expression. Cystatin C expression was detected in 6 of 18 samples which were further characterized by absent (5 cases) or only residual (1 case) trypsin activity. Trypsin and cathepsin B activities were observed in 8-10 samples and clearly showed a positive correlation. In spite of the heterogenous activation status of the pancreatic juice samples, correlation analysis of CST3 densitometry and trypsin activity showed a significant negative correlation (Spearman $r=-0.6078$ with a p value of 0.0075) which suggests a degradative effect of

trypsin or trypsin-activated zymogens on CST3. Beside trypsin all other digestive enzymes secreted by the pancreas are present in pancreatic juice, such as elastases and carboxypeptidases which were able to degrade proteins in total. We included these new data in the manuscript as supplemental figure S1: “In 18 additional samples of human pancreatic juice we could demonstrate a negative correlation of the presence of CST3 with CTSB and trypsin activity (S1 A-F).”

Fig. S1: (A) Densitometric analysis of CST3 Western-blot expression in human pancreatic juice (n=18). (B and C) Trypsin and cathepsin B activity measurements in these samples (dots reflect technical replicates of each sample). (D) Cathepsin B activity showed a positive correlation with trypsin activity (Spearman $r=0.711$, $p=0.0009$, $n=18$). (E and F) Densitometric analysis of CST3 expression in human pancreatic juice showed a negative correlation with (E) cathepsin B activity (Spearman $r=-0.7606$, $p=0.0002$, $n=18$) as well as trypsin activity (F) (Spearman $r=-0.6078$, $p=0.0075$, $n=18$).

3. Line 138: “we measured enzyme activation” instead of “we measured enzyme activities” would be more correct as enzyme activities over time are presented in Figure 2E.

Response: Actually, the figure shows a kinetic of the enzymatic activities of cathepsin B and trypsin in the first 60 min after pancreatitis induction. Enzyme activities were measured by the cleavage of a specific substrate which emits a fluorescent signal. The signal strength reflects the amount of active enzyme at the moment of the measurement and includes enzyme activation, but also enzyme degradation events. As the figure does not show the accumulated “activation” we think that “activities” is the more correct term.

4. Figures 2B and 2D show enzyme activities in units RFU/mg and RFU/ μ g. It seems that the time component is missing from the figures (RFU/min/mg and RFU/min/ μ g). It is not clear why the activity values for trypsin and chymotrypsin are largely different by comparing Figures 2B and 2D.

Response: We apologise for that issue and added the time component to the axis label. In Figure 2B we have measured the trypsin and chymotrypsin activities after enterokinase activation, which reflects the total (hypothetic) trypsin and chymotrypsin content of wild type and *Cst3*^{-/-} mice after complete activation of trypsinogen. During pancreatitis or after stimulation of isolated acinar cells with 0.001mM CCK (like in Fig 2D) not all trypsinogen is converted to trypsin, and only a minor percentage of zymogens becomes activated. This small amount is still enough to damage the cells and induce pancreatitis. This is the reason why the axis scale is significantly different between the Fig. 2B and 2D.

5. The use of chronic pancreatitis patients for the genetic studies should be clarified, given that CST3 deficiency is associated with acute pancreatitis.

Response: Acute and chronic pancreatitis can be seen as two faces of the same underlying disease and in most cases chronic pancreatitis starts with a first episode of acute pancreatitis. The most frequent precipitating factors of AP are gall stones and environmental factors like alcohol and tobacco, but also some genetic risk factors have been identified. The genetic risk factors play a much bigger role in patients with recurrent or chronic disease where they are much more frequently detected. Thus, a genetic study in CP patients is much more likely to identify a potential role of CST3 variants in pancreatitis. We included the following sentence into the manuscript to explain the genetic analysis of CP patients: “Due to a higher prevalence of genetic risk factors in patients

suffering from recurrent or chronic disease we analysed the potential role of CST3 variants in CP patients.”

6. Does CST3 contain any disulfide bonds? Disulfide bonds may prevent the degradation of the inhibitor, even if it is cleaved.

Response: Yes, murine CST3 contains two intramolecular disulfide bonds between the amino acids CYS93-CYS103 and CYS117-CYS137 (see figure below). Moreover, we used Nano differential scanning fluorimetry to determine the protein stability of full-length cystatin C and truncated cystatin C. While full-length cystatin C had a melting point of 81.0 °C, a comparable melting point of trypsin cleaved cystatin C was 80.6 °C. Therefore, it can be concluded that full-length as well as cleaved cystatin C are very stable and that trypsin cleavage does not impair the protein stability. We include this data in the manuscript: “The CST3 molecule is stabilized by two disulfide bonds and the protein stability is not affected by trypsin cleavage (**Fig. S3**).”

Fig. S3: AlphaFold2 structure of murine cystatin C with highlighted cysteine residues that form disulfide bridges.

7. Protease activities were measured for 60 min at 37 °C. How the evaporation of the samples was prevented? Was the kinetic linear over the course of these long measurements?

Response: Enzyme kinetics are measured in 96-well plates which are covered by a lid. Evaporation over 60 min at 37°C from a volume of ~100µl is hardly measurable and would affect samples and controls in similar ways. We could not observe a significant reduction of the volume over the period.

The enzyme kinetics of trypsin, chymotrypsin, cathepsin B and cathepsin L were determined fluorometrically over 60 min in triplicate for each individual sample. The enzyme kinetics behave differently for the respective protease and the corresponding substrate. While trypsin and

cathepsin B were mostly in the linear range until the end of the measurement time, chymotrypsin or cathepsin L measurements reached saturation in the last few minutes especially for the 1 h pancreatitis samples. All samples were measured over 60 min, but only a time window in which all samples were in the linear range of the respective enzyme was used to analyse and determine the enzyme activity. The time windows were chosen to be the same for all samples for the respective enzyme to ensure comparability (red window).

8. Line 217: The abbreviation MDS is unnecessary as it appears only once in the text. Line 366: Sentence starting with “Trypsinogen T7 deficient mice...” is incomplete. Words are missing from the end of the sentence.

Response: We deleted the abbreviation MDS in line 217 and added the missing word “differences” in line 366.

9. If independent samples were measured, using the mean \pm SD would be more appropriate than mean \pm SEM.

Response: As the reviewer suggested, we changed all graphs in the manuscript from mean+SEM to mean+SD.

10. The manuscript contains some typos such as promoter (line 294), with (line 415).

Response: We corrected the indicated typos.

Reviewer #2 (Remarks to the Author):

This manuscript investigates the role of cystatin C during acute pancreatitis and suggests a novel allosteric interaction with cathepsin B. Overall the results are interesting and important, but addressing the below queries would strengthen the conclusions.

-Figure 1B – Please provide the whole blot for cathepsin B, which includes the size of the pro-form, single and heavy chain. It would also be helpful to see a blot for cathepsin L, given differences in activity shown in 1D. What is the purpose of the dashed line in the gel? It gives the impression that the gel is cropped, but if this is not the case, suggest removing it.

Response: As the reviewer suggested we included a Western-blot where also the pro-form of cathepsin B is visible. The dashed line had been introduced to separate the 0 h and the 1 h samples. The blot in the figure shows one gel, so we removed the line. We also included a Western blot for cathepsin L in the figure.

-Line 119 and Figure 1H – the text states pancreatic juice from patients. Please specify what patients? Do they have pancreatitis? No information about these samples is provided in the methods section.

Response: We apologise that we had not included the patient information in the manuscript. We now implemented the pancreatic juice collection in the materials and methods section. Pancreatic juice of patients, suffering from pancreatic cancer as well as from chronic or acute pancreatitis, was collected during and after surgical procedures. Because of the interventional treatment of the patients and different lengthy freezer storage periods after collection some samples could contain secreted zymogens which are autoactivated and can degrade the proteins in the sample. We added the following paragraph to the material and method section of the revised manuscript:

“Human pancreatic juice samples

Pancreatic fluid was collected within the GPCN study (Reg.-Nr: III UV 05/06) during or after interventional or surgical treatment of patients with pancreatic disease. All individuals gave their written informed consent for participation and the local ethics committee approved the study protocol. The 18 samples include 11 patients with chronic pancreatitis, 6 patients with pancreatic cancer and 1 patient suffering from acute pancreatitis.”

-In Figure 2F, is there a significant difference between cathepsin B activity in the ZG of WT mice between untreated and pancreatitis conditions? If so, this should be indicated. This same comparison is shown to be significant in 1C, but it’s not clear if this is consistent given the spread of the data. In Fig 3A, there is no difference at pH 5.5 (the pH of 1C and 2F). Why is it inconsistent?

Response: The apparent redistribution of cathepsin B activity from the lysosomal compartment to the secretory compartment during the course of acute pancreatitis, as shown in Figure 1C, is an established phenomenon that has been previously reported in different animal models of acute pancreatitis [1-3]. In Figure 2F we observed increased cathepsin B activity in the ZG fraction of wt mice at 1 h of pancreatitis, but this increase did not reach statistical significance in this measurement. In Figure 3A and B we investigated the pH-optimum of cathepsin B activity under normal and pancreatitis conditions. The graph shows a maximal “relative” activity of CTSB at pH 5.5 (100%=1) in control zymogen granules and the ZG fraction 1 h after induction of pancreatitis by caerulein, but a further increased activity of cathepsin B is seen during pancreatitis at a slightly higher pH of 6. So, the graph actually shows a higher maximal CTSB activity under slightly higher pH conditions during pancreatitis.

1. Saluja A, Hashimoto S, Saluja M, Powers RE, Meldolesi J, Steer ML. Subcellular redistribution of lysosomal enzymes during caerulein-induced pancreatitis *Am J Physiol.* 1987 Oct;253(4 Pt 1):G508-16.
2. Saluja A, Saluja M, Villa A, Leli U, Rutledge P, Meldolesi J, Steer M. Pancreatic duct obstruction in rabbits causes digestive zymogen and lysosomal enzyme colocalization *J Clin Invest.* 1989 Oct;84(4):1260-6.

3. Saluja M, Saluja A, Lerch MM, Steer ML. A plasma protease which is expressed during supramaximal stimulation causes in vitro subcellular redistribution of lysosomal enzymes in rat exocrine pancreas *J Clin Invest.* 1991 Apr;87(4):1280-5.

-If trypsin and chymotrypsin are driving acinar death, then why is there no difference in necrosis between WT and *Cst3^{-/-}*, where there is significantly more activation of these enzymes?

Response: This is a very good question, but we think that the reason is the use of a hormone-hyperstimulation model for the induction of pancreatitis. Caerulein is a Cholecystokinin (CCK) receptor agonist. Caerulein- as well as CCK-induced pancreatitis are standardized disease models of acute pancreatitis, and both cause a prolonged intracellular Ca^{2+} release in pancreatic acinar cells [1]. While oscillating Ca^{2+} signals result in the secretion of the zymogen-containing granules at the apical pole of the cell, a prolonged cytosolic Ca^{2+} release leads to a secretion blockade which is associated with an intracellular activation of trypsinogen [1]. In addition to the protease activation, the pathological Ca^{2+} signal also induces other intracellular signalling pathways independent of trypsinogen activation, such as the activation of NFkB [2] or a depolarisation of the mitochondria [3]. Therefore, although a difference in trypsinogen activation can be measured, this does not necessarily correlate with the necrosis rate, which is triggered also by other calcium-dependent mechanisms.

1. Krüger B, Albrecht E, Lerch MM. The role of intracellular calcium signaling in premature protease activation and the onset of pancreatitis *Am J Pathol.* 2000 Jul;157(1):43-50.
2. Gukovsky I, Gukovskaya AS, Blinman TA, Zaninovic V, Pandol SJ. Early NF-kappaB activation is associated with hormone-induced pancreatitis. *Am J Physiol.* 1998 Dec;275(6):G1402-14.
3. Mukherjee R, Mareninova OA, Odinkova IV, Huang W, Murphy J, Chvanov M, Javed MA, Wen L, Booth DM, Cane MC, Awais M, Gavillet B, Pruss RM, Schaller S, Molkenin JD, Tepikin AV, Petersen OH, Pandol SJ, Gukovsky I, Criddle DN, Gukovskaya AS, Sutton R; NIHR Pancreas Biomedical Research Unit. Mechanism of mitochondrial permeability transition pore induction and damage in the pancreas: inhibition prevents acute pancreatitis by protecting production of ATP *Gut.* 2016 Aug;65(8):1333-46.

-What is the relationship between cathepsin B/L and chymotrypsin? There is a much bigger difference in chymotrypsin activation in the absence of cystatin C compared to trypsin. It would thus be critical to show whether chymotrypsin can cleave cystatin C.

Response: As suggested by the reviewer we performed degradation experiments of cystatin C by chymotrypsin. We did not observe a degradation of cystatin C by chymotrypsin, as western blot analysis showed no size shift of cystatin C even after 60 min of co-incubation with chymotrypsin. Furthermore, even after 60 min of co-incubation cystatin C showed the same inhibitory capacity against CTSB and CTSL. Of interest we observed a reduced activity of both cathepsin B and L in the presence of chymotrypsin alone, which suggests a degradative effect of chymotrypsin on CTSB and CTSL. We included these data as supplementary figure 2D-F in the revised manuscript.

Fig S2: (D) Western blot analysis of the time-dependent cleavage of cystatin C by α -chymotrypsin. **(E)** Measurement of CTSB activity at pH 5.5 in the presence of cystatin C, α -chymotrypsin and preincubated cystatin C with α -chymotrypsin, **(F)** measurement of CTSL activity at pH 4.0 in the presence of cystatin C, α -chymotrypsin and preincubated cystatin C with α -chymotrypsin (n=4).

-Fig 3G-H, please indicate the assay pH in the figure legend (this would be helpful to include throughout).

Response: We included the assay pH in the figure legend.

-Fig 3A-D Please indicate in the figure legend what the red line on the graphs represents. It is not clear – presumably the pH optimum for each enzyme? If so, it is in the wrong spot for cathepsin B (shown to be 5.5 in Fig S1).

Response: To avoid any misinterpretation, we removed the red lines that indicated the pH value showing the maximal activity (= 1.0), which was used to calculate the activity ratios at other pH values.

-Also in Figure S1, why does cathepsin L have no activity at pH 6.5 but it's active at 7? If there were no data points for 6.5, this should be indicated to avoid confusion. Error bars for this graph would also be helpful.

Response: This data point had not been measured. We have replaced the graph and added standard deviation +/-SD to all data points. As suggested, we mention that the activity was not measured when the data point is missing.

Fig. S2: (A) The comparison of pH-related enzyme activities of CTSB and CTSL in ZG fractions (n=4) showed a pH optimum of CTSL at pH 4.5 and of CTSB at pH 5.5. Dots represent measurements +/- SD (CTSB activity was not measured at pH 2.0 and 3.0 whereas CTSL activity was not measured at pH 6.5 and 8.0).

-Given the R28 truncation occludes the essential cathepsin B-interacting residues, how does the trypsin-cleaved version (which is primarily cleaved at this site, along with K24) enhance cathepsin B activity? It would be important to show experimentally that the shorter truncations promote activation even further (e.g., R71, 73, 74).

Response: We added additional experiments to show the influence of pH shifts and the truncation by trypsin on the dimerization of CST3 (Fig. 4H-I). We can demonstrate that both pH drop and trypsin truncation induce the dimerization of CST3. This can be explained by an easier release of the N-terminus from its locked position at the beta-sheet and the subsequent opening of the sandwich structure, which is formed from the helical and sheet motifs. Since pH changes accompany the onset of AP and a subsequent trypsin activation, we assume that these conditions contribute to the release of the N-terminus of CST3 and allow its allosteric binding and modulation of CTSB. We discussed this point in the manuscript and added the following sentence: "As it is assumed that pH shifts trigger the onset of AP, and as we demonstrated that the dimerization of CST3 is dependent on both the pH change and the truncation by trypsin (Fig. 4I-H), the release of the CST3 N-terminus must be favoured by these conditions, too."

-Does the truncation somehow promote dimerization? That seems like what the model is suggesting, but there is no evidence for this.

Response: We performed additional experiments where we rebuffered full-length cystatin C and trypsin cleaved cystatin C in buffers with different pH values. After storage the monomer/dimer ratios were determined using size exclusion chromatography. We included these additional data in

one figure (Fig. 4H-I) and in the supplementary information (Fig. S4, Fig. S5). To our surprise, the trypsin truncation also induced CST3 dimerization.

We added the following sentences to the results section: “We further investigated the pH-dependent dimerization of cystatin C and could show that at pH 4.0 100 % of cystatin C was present as a dimer. Between pH 5.0 and 5.5 we observed a switch in the ratio of predominant dimeric to more monomeric forms of cystatin C, which persisted at higher pH values (Fig. 4H, Fig. S4). These results fit to the previously reported observation that CST3 forms more dimers under acidic conditions [1]. Moreover, we analyzed if the trypsin mediated cystatin C truncation also influences the monomer/dimer ratio. We found that at pH 4.0, 100 % dimer was present as it was also shown for the full-length CST3. However, dimers remained as the predominant (~70%) form at pH values from 5.5 to 8.0 (Fig. 4I, Fig. S5). We conclude that the trypsin mediated truncation of cystatin C, indeed, further favours the dimerization of CST3.”

1. Ekiel, I. & Abrahamson, M. Folding-related dimerization of human Cystatin C. *J. Biol. Chem.* 271, 1314–1321 (1996).

Fig. 4: (H-I) Comparison of pH related dimerization of full-length cystatin C (H) and trypsin cleaved cystatin C (I). Shown are the dimer/monomer ratios obtained in sodium acetate buffer pH 4.0 to 5.5 (dark blue and red, respectively) and the dimer/monomer ratios obtained in sodium phosphate buffer pH 5.5 to 8.0 (light blue and orange, respectively).

Fig. S4: Chromatograms of the separation of cystatin C monomer and dimer fractions by size exclusion chromatography. (A-C) Cystatin C solutions stored and separated in sodium acetate buffer (100 mM sodium acetate, 150 mM sodium chloride) at pH 4.0 (A), pH 5.0 (B) and pH 5.5 (C). (D-G) Cystatin C solutions stored and separated in sodium phosphate buffer (100 mM sodium phosphate, 150 mM sodium chloride) at pH 5.5 (D), pH 6.0 (E), pH 7.0 (F) and pH 8.0 (G).

Fig. S5: Chromatograms of the separation of trypsin-mediated cleaved cystatin C monomer and dimer fractions by size exclusion chromatography. (A-C) Truncated cystatin C solutions stored and separated in sodium acetate buffer (100 mM sodium acetate, 150 mM sodium chloride) at pH 4.0 (A), pH 5.0 (B) and pH 5.5 (C). (D-G) Truncated cystatin C solutions stored and separated in sodium

phosphate buffer (100 mM sodium phosphate, 150 mM sodium chloride) at pH 5.5 (D), pH 6.0 (E), pH 7.0 (F) and pH 8.0 (G).

Experimental details were also added to the method section:

“pH-dependent dimerization of cystatin C and trypsin cleaved cystatin C

After the purification step or trypsin cleavage, the same amounts of cystatin C or truncated cystatin C, originating from the same expression batch, were rebuffed in seven different buffers over a pH range from 4.0 to 8.0. The used buffers were sodium acetate buffers (100 mM sodium acetate, 150 mM sodium chloride, pH 4.0, 5.0 and 5.5, respectively) and sodium phosphate buffers (100 mM sodium phosphate, 150 mM sodium chloride, pH 5.5, 6.0, 7.0 and 8.0, respectively). The protein solutions were stored at 4 °C in the corresponding buffer overnight. On the next day, the ratio of monomer and dimer fractions of the 3 mg/mL protein solutions in the different buffers was analysed using size exclusion chromatography.”

“Digestion of cystatin C by trypsin and chymotrypsin

To identify the monomer/dimer-ratio of trypsin-digested cystatin C, after purification 14 mL of a 2 mg/mL cystatin C solution was rebuffed in trypsin buffer (100 mM Tris, 5 mM CaCl₂, pH 8.0). The reaction was started by the addition of 60 µl of a 25 µg/mL trypsin solution and lasted 13 hours at 37 °C at 180 rpm. Full cystatin C truncation by trypsin was verified by SDS-PAGE analysis on the next day.”

-What proportion of cystatin C is in monomer v dimer form in the pancreas? It would be important to show that this balance is skewed towards dimer after pancreatitis initiation. Moreover, experimental evidence that cystatin C is actually truncated *in vivo* during pancreatitis is lacking. If this is a physiologically relevant mechanism, would you not expect to see cleaved cystatin C in the western blot shown in 1B? There doesn't appear to be, or at least not as dramatic as observed in the cleavage assays. There may be some evidence of cleavage in the human pancreatic juice, but it's not clear that this is from patients with pancreatitis, and it is extra-granular so may not represent what is actually happening inside the pancreas. Proteomics/N-terminomics on ZG proteins with/without caerulein would reveal the proportion of cleaved/uncleaved cystatin C (if any).

Response: The question of whether dimer/monomer formation can be observed *in vivo* is difficult to answer. The cathepsin B-mediated activation of trypsinogen to trypsin takes place in the heavy compartment (secretory vesicles), but not all zymogen vesicles react simultaneously in a synchronized way, so that only a small percentage of trypsinogen becomes activated. V-ATPase-mediated acidification triggers the activation [1] and as we have shown, the pH is decisive for the dimerization of cystatin C. The detection of dimerisation *in vivo* would be a challenge, because it is

not possible to mimic the intravesicular pH which defines the dimer/monomer ratio. *Ex vivo* it is the buffer conditions which define the pH and therefore the monomer/dimer ratio. In native gels we would have the same artificial buffer conditions for all samples and therefore could not observe any differences. One way to investigate the influence of dimers on CTSB activity would be to use a transgenic animal model containing a mutated dimer binding pocket in the CTSB, but this is not available.

Also the reviewer points out that there is no prominent cleaved CST3 in W-blot as in the cleavage assays. We performed additional western blot analysis of zymogen granules using 17.5% polyacrylamide gels to analyse degradation events. In the zymogen granule fraction of wild type mice 1 h after onset of pancreatitis we could observe a degradation of CST3 in the ZG fraction. Densitometry analysis of CST3 showed a significant reduction of full length CST3, whereas a smaller form of CST3 could be detected (figure for reviewer).

Figure for the reviewer: Densitometric analysis of CST3 western blot of pancreatic zymogen granule fraction of control mice (0 h) and mice with pancreatitis (1 h).

The observed shift suggests a cleavage/degradation of CST3 under pancreatitis conditions. Trypsin mediated processing of cystatin C occurs within minutes and continues during our enzyme activity measurements. Indirect evidence for the degradation of cystatin C is also seen by the shift of the pH optimum of cathepsin B in the ZG fraction (Fig. 3A). 1 h after the onset of pancreatitis, the pH curve for cathepsin B activity is shifted, and exactly parallels the pH curve of CTSB activity in *Cst3*^{-/-} animals. This strongly suggests that cystatin C can no longer inhibit cathepsin B. Furthermore, the studies in animals suggest a protective role of *Cst3* during pancreatitis, as disease severity and trypsinogen activation are significantly increased in these mice. Proteomics/N-terminomics struggle with the problem of protease contamination by various pancreatic proteases including trypsin, elastases and carboxypeptidases which does not allow to identify a specific N-terminal cleavage site from one of these enzymes.

Fig. S2: (B) Western Blot of zymogen granule fractions of control mice (0h) and of mice treated with caerulein for 1h. Each lane represents an independent sample. **(C)** Cystatin C was detected by using anti-cystatin C antibody, densitometry showed the ratio of uncut CST3 to processed CST3 (n=5). Significance was calculated by two-tailed Student t test for independent samples. Results are shown as mean \pm SD. Significance levels of $p < 0.05$ are marked by an asterisk.

The discussion indicates that low pH induces cystatin C dimerization. The main differences in cathepsin B activity upon caerulein treatment were observed above pH 6.5. In that case, it would be important to look at dimerization of the cleaved cystatin 3 across a wide pH range.

Response: We had performed additional experiments as mentioned above, which clearly show that low pH induces cystatin C dimerization (**Fig. 4H-I, Fig. S4, Fig. S5**). Moreover, we could show that the monomer/dimer ratio of cystatin C is shifted in favour of dimers, when truncated by trypsin.

-The potential allosteric site within cathepsin B is very interesting, but whether or not it is relevant to pancreatitis is still unclear. These data might be more compelling as two separate manuscripts, where the novelty of the interaction between cystatin c and cathepsins is the focus.

Response: This is a valid point of discussion. The mechanism shown here was discovered during experiments investigating the activation of trypsinogen in the secretory compartment of acinar cells. We hypothesised first that a degradation of cystatin C in the secretory compartment allows cathepsin B to activate trypsinogen. We did see a degradation (**Fig. 3E**), but surprisingly, the degradation product showed an activity-increasing effect on cathepsin B. Based on this observation, we identified the second binding pocket in cathepsin B. The dimerization of CST3 is both pH-dependent and enhanced by tryptic cleavage. Both processes are known to be relevant for the onset of acute pancreatitis, which suggests that this mechanism could play a role in the onset of pancreatitis. Furthermore, trypsinogen is only expressed in pancreatic acinar cells, where it is activated intracellularly to trypsin during pancreatitis. ZG are the only compartment where trypsin, cathepsin B and cystatin C are co-localized. For this reason, we have decided to publish the data in

one manuscript, as it provides an explanation for the initial events during pancreatitis in addition to the biochemical aspect.

-The occluding loop of cathepsin B is important for governing its endo and exopeptidase activity (which is pH dependent). It would be worth discussing how cystatin C binding may shift the substrate repertoire.

Response: The occluding loop is a very flexible loop where the conformational change of this loop is responsible if the endo- or exopeptidase activity of cathepsin B is carried out. The substrate Abz-GIVRAK(Dnp)-OH was commonly used to assay the dipeptidyl carboxypeptidase activity of cathepsin B with an optimum at acidic pH of 4.5 to 5.5 [1]. Z-Arg-Arg-AMC is widely used as the endopeptidase substrate with a pH optimum between pH 6.0 and neutral pH. From these results it has been proposed that the dipeptidyl carboxypeptidase activity of cathepsin B is favoured at lower pH values while the endopeptidase activity is enhanced at neutral pH. However, in 2022 it was shown using a peptide library that the activity differences were rather due to the used substrates than the pH. For example, it was identified that if endopeptidase activity at lower pH values can be measured with the endopeptidase substrates Z-VR-AMC and Z-ER-AMC [2]. This was evidence that the endopeptidase and exopeptidase activity both occur under neutral and acidic conditions. More important, in general, is the formation of the occluding loop. It is known that the two amino acids His110 and Arg116 present in the occluding loop are able to form two salt bridges with the amino acids Asp22 and Asp224 to create a compact structure [3]. The presence of the occluding loop when it's close to the active site in the compact structure favours the dipeptidyl carboxypeptidase activity. This is mainly due to the two positively charged amino acids in the occluding loop His110 and His111 that can bind the C-terminal carboxylate group of substrates. Using site-directed mutagenesis it was shown that CTSB possesses higher endopeptidase activity when the salt bridges of the loop and the other parts of CTSB are broken [3]. We had shown in the manuscript that due to the dimer binding of cystatin C to CTSB there is a conformational change which removes the occluding loop from the vicinity close to the active site and therefore favours the endopeptidase activity.

We added the following sentences to the discussion section: "CTSB is known to have endo- and exopeptidase activity depending on the conformation of the occluding loop. If the occluding loop is in close vicinity to the active site and therefore CTSB forms a rather closed and compact structure, the exopeptidase activity is favoured, where CTSB removes dipeptides from the C-terminus of proteins or peptides. If the occluding loop moves away from the active site the endopeptidase activity is increased [3]."

1. Cotrin, S. S.; Puzer, L.; de Souza Judice, W. A.; Juliano, L.; Carmona, A. K.; Juliano, M. A. Positional-scanning combinatorial libraries of fluorescence resonance energy transfer

peptides to define substrate specificity of carboxydipeptidases: Assays with human cathepsin B. *Anal. Biochem.* 2004

2. Yoon MC, Hook V, O'Donoghue AJ. Cathepsin B Dipeptidyl Carboxypeptidase and Endopeptidase Activities Demonstrated across a Broad pH Range *Biochemistry.* 2022 Sep 6;61(17):1904-1914.
3. Nägler DK, Storer AC, Portaro FC, Carmona E, Juliano L, Ménard R. Major Increase in Endopeptidase Activity of Human Cathepsin B upon Removal of Occluding Loop Contacts *Biochemistry.* 1997 Oct 14;36(41):12608-15.

-Cystatin C can also inhibit legumain through a distinct binding site. It would be important to mention this and discuss potential implications.

Response: The cysteine protease legumain has a strict specificity for hydrolysis of asparaginyl bonds and may be involved in the processing of bacterial peptides and endogenous proteins for MHC class II presentation in the lysosomal/endosomal systems. Enzyme activation is triggered by acidic pH and appears to be autocatalytic. Protein expression occurs after monocytes differentiate into dendritic cells. A fully mature, active enzyme is produced following lipopolysaccharide expression in mature dendritic cells. However, analysis of spatial expression patterns across organs and tissues by RNA-Seq revealed the lowest expression of legumain in human pancreas, compared to 22 other tissues [1].

1. Fagerberg L, Hallström BM, Oksvold P, Kampf C, Djureinovic D, Odeberg J, Habuka M, Tahmasebpoor S, Danielsson A, Edlund K, Asplund A, Sjöstedt E, Lundberg E, Szgyarto CA, Skogs M, Takanen JO, Berling H, Tegel H, Mulder J, Nilsson P, Schwenk JM, Lindskog C, Danielsson F, Mardinoglu A, Sivertsson A, von Feilitzen K, Forsberg M, Zwahlen M, Olsson I, Navani S, Huss M, Nielsen J, Ponten F, Uhlén M. Analysis of the human tissue-specific expression by genome-wide integration of transcriptomics and antibody-based proteomics *Mol Cell Proteomics.* 2014 Feb;13(2):397-406.

-CSTB is often used instead of CTSB (CSTB is stefin B). CTS3 is used in place of CST3. Cys3^{-/-} is used in some of the figures instead of Cst3 (e.g. 3B, D). Please amend.

Response: We corrected these issues in the revised manuscript.

-Sometimes bar graphs show only error bars, while others show the mean, SEM, and the individual data points. The latter is much more useful, so it would be helpful to use that style consistently.

Response: As suggested by reviewers 1 and 2, we replaced all graphs by new graphs containing individual points and +/-SD instead of SEM.

Reviewer #3 (Remarks to the Author):

In a study entitled „Biochemical Analyses of Cystatin-C Dimers and Cathepsin-B reveals a Trypsin-Driven Feedback Mechanism in Acute Pancreatitis“ the authors provide compelling evidence that co-localization of the cysteine protease CTSB with trypsinogen within the zymogen-containing secretory

compartment is not sufficient for trypsinogen activation since that under physiological conditions CTSB activity is regulated by the type 2 cystatin protein inhibitor, CST3, making it a key enzyme in the development of pancreatitis. Using computational methods, the authors provided an understanding of the autoinhibitory feature of CTSB and the mechanism by which the CST3 monomer effectively supports this autoinhibition by preventing exposure of the active site. Furthermore, the dimeric form of CST3, together with tryptic degradation products, promotes conversion to trypsin by binding to the allosteric binding site and stabilizing the occluding loop in an open conformation, thus making the active site more receptive to trypsinogen binding.

Although the performed MD simulations and the way the authors presented the results reliably support the experimental observations, I have a few questions regarding the computational part:

1) Is there a particular reason why the modelling was done on murine rather than human proteins, especially if human crystal structures are available? What is the percentage of sequence identity between human and murine CTSB and CST3?

Response: The modelling was primarily done on murine proteins due to the fact that the initial observations of the described effects have been received from murine proteins in a murine experimental pancreatitis model. Due to the limited access to human material, murine disease models are the only possibility to perform pathophysiological studies on pancreatitis. For this reason, we decided to carry out the experiments with murine proteins in order to compare them with the animal experimental data. During biochemical analysis of the increased CTSB activity we used isolated murine and human proteins also across species and observed sometimes even stronger effects without prior TRY1 cleavage of CST3. We therefore investigated different mono-/dimer ratios within protein batches from different sources. As a result, the increased CTSB activity was attributable to the dimerisation state of CST3 and was not species-specific. Sequence similarities are mentioned in the text: "In the human systems CTSB, CTSL and CST3 have high protein sequence homologies of 79.5%, 72.1% and 69.2% with the murine counterparts".

2) Did the authors generate the initial CTSB:CST3 complex using AF2 or did they model the proteins individually and then obtain the complex by aligning the structures with that of PDB 1STF?

Response: Yes, the proteins were modelled individually and then aligned with PDB 1STF. This has also been mentioned in the methods section. Subsequently, binding poses within the active site and the involvement of the flexible N-terminal were sampled with sophisticated replica-exchange MD simulations.

3) Is there any particular reason why restraints were applied on the all backbone atoms (except the occlusion loop of CTSB)? Could this have an effect on the assessment of allosteric interactions?

Response: These restraints are generally meant to prevent too large structural changes due to extended time periods at elevated temperatures during TIGER2h simulations, and restrain the accessible structural ensemble within a reasonable range. However, this is more of a safety precaution for the efficiency of the simulation (not losing replicas to sample unimportant structural spaces), rather than an actual influence during the simulation. Because these restraints are implemented in a flat-bottom fashion with a generous upper threshold value for the RMS deviation, they will only counteract mature denaturation events when the RMSD exceeds the threshold, but will not be active below. Hence, most of the simulation time, no force is applied at all (see plot below). Since the occluding loop was expected to be rather flexible, it was excluded from these restraints, which are to detect these mature denaturation events, rather than exerting an influence on functional flexibility. We therefore do not expect an influence of these restraints on the modulation of the occluding loop.

4) When authors say that the occluding loop has shifted from its usual position upon binding of monomeric CST3, does this mean that the active site is still protected, but this time by the CST3 N-terminus?

Response: Yes, the active site is now protected by inhibitory binding loops of CST3, while the occluding loop is slightly shifted away from its position, to make room for CST3, comparable to induced fit binding.

5) Could the authors show the interaction network of CTSL and CST3 during inhibition for the dominant conformation (Cluster 2), instead of Cluster 4, since according to Fig. S3 (E) K24 appears to be quite important for CTSL binding, while R28 also established a contact for more than 25% of the simulation time? In comparison, in the CST3:CTSB simulation in Cluster 0, K24 does not form interactions that persistent more than 25% of the time (Fig. S2 (F)). What suggests that the N-terminal region of CST3 is not as important for CTSL binding compared to CTSB?

Response: We decided to show the network for Cluster4, because it was more similar to the binding pose obtained for CTSB. Cluster4 featured a unique binding position and tight locking of the K24 and R28 residues, while the N-terminus shows much greater flexibility and no distinct binding mode in Cluster2 (see plot below for RMSD of K24/R28 in both clusters). As rightfully mentioned by the reviewer, the dominant conformation was Cluster2 and the lack of a distinct binding position for the N-terminus in this cluster may explain why the TRY1 cleavage does not weaken the inhibition for CTSL as strongly as for CTSB, given that this is the mature binding mode. We refined this discussion in the text. Please find below the contact network for cluster2, which still shows involvement of K24 and R28, but as shown by the RSMD graph, they do not show a distinct binding pose.

In Fig. S3 (E), CTSL should be written instead of CTSB.

Response: This has been corrected.

6) It is not clear to me why CST3 fragments cleaved at position R71 and R45 were computationally studied if K24 and R28 were found to be tryptic cleavage sites?

Response: Theoretical investigations accompanied over several years the extensive experimental work of this study. During this time, the correct TRY1 cleavage sites at CST3 were elucidated step by step and required multiple different attempts. Initial mass spectrometry indicated possible cleavage sites at R71 and R45, which were used during the first MD simulations and initially identified the allosteric pocket. By further experimental efforts the cleavage site was later narrowed down to the final K24/R28 sites in the dimeric complex. New insights were repeatedly adapted in lengthy computational model simulations, leading to the series of CST3 cleavage constructs that we investigated.

7) How the initial complexes of the processed m-/h- monomeric/dimeric CST3 with CTSB were generated – by docking, by using AF2?

Response: The CST3 cleavage construct R71 was first situated close to the active site of CTSB, leading to no favourable binding structures. A second attempt placed CST3-R71 on the opposite site of CTSB and returned the binding pose within the newly discovered allosteric pocket of CTSB. Subsequent iteration rounds, on updated models of the CST3 cleavage constructs, always used a prepositioned structure close to this allosteric pocket, to check for reproducibility of the occluding loop opening.

8) When human CST3 in dimer form was simulated, was it simulated in complex with human or murine CTSB (Fig. 6 (A))?

Response: The human dimeric CST3 construct was simulated with murine CTSB, because intermediate results from experimental activity measurements (data not shown) with human CST3 revealed a strong over activation effect on murine CTSB, even without TRY1 processing. We later discovered that the mono-/dimer ratio in the human CST3 sample was tilted much more towards dimers. This is, however, rather a consequence of the protein production (supplier, own purifications), rather than a consequence of the species. During our trials with human and murine protein samples, no strong influence of the species of origin could be determined, neither on inhibition nor on activation. Apparently, this interaction represents a well conserved mechanism.

9) Have the authors tried to model tryptic CST3 fragments with CTSL as well? Do they bind in the allosteric site or the orthosteric site was retained?

Response: No, we did not model CTSL with tryptic cleavage constructs of CST3. This allosteric modulation seems to be special to CTSB. While the core structure of CTSL is very similar, CTSL does not feature a motif like the occluding loop, nor has it a disordered region to connect and build the allosteric site, like in CTSB. Also, in experimental activity measurements, following the same

methodology used with CTSB no such influence could be found. Instead, CTSL is still inhibited by TRY1 processed CST3, and no allosteric modulation towards higher activity was detected. (s. Fig. 4G).